# ON LIMITATION OF TRANSFORMER FOR LEARNING HMMS

## ABSTRACT

This paper investigate the capability of transformer in learning a fundamental sequential model — the Hidden Markov Model (HMM). We design various types of HMM examples and variants inspired by theory, and conduct extensive experiments testing and comparing the performance of both transformers and Recurrent Neural Networks (RNNs). Our experiments reveal three important findings: (1) Transformers can effectively learn a large number of HMMs, but this require the depth of transformers to be at least logarithmic in the sequence length; (2) There are challenging HMMs where Transformers struggle to learn, while RNNs succeed. We also consistently observe that Transformers underperform RNNs in both training speed and testing accuracy across all tested HMM models. (3) Long mixing times and the lack of access to intermediate latent states significantly degrade Transformer's performance, but has much less impact on RNNs' performance. To address the limitation of transformers in modeling HMMs, we demonstrate that a variant of the Chain-of-Thought (CoT), called *block CoT* in the training phase, can help transformers to reduce the evaluation error and to learn longer sequences at a cost of increasing the training time. Finally, we complement our empirical findings by theoretical results proving the expressiveness of transformers in approximating HMMs with logarithmic depth.

## 1 INTRODUCTION

Transformer-based architectures (Vaswani et al., 2017) have demonstrated exceptional capabilities in tackling sequential modeling tasks across diverse domains, including natural language processing (Brown et al., 2020), computer vision (Dosovitskiy et al., 2020), robotics (Brohan et al., 2023), reinforcement learning (Janner et al., 2021; Lee et al., 2022), etc. Despite their widespread success, the effectiveness of Transformers in learning basic sequential models, such as the Hidden Markov Model (HMM), remains unclear. Investigating this question is crucial for understanding the strengths and limitations of Transformers, especially considering that HMMs are arguably among the simplest yet fundamental tools for modeling natural language (Merialdo, 1994; Vogel et al., 1996; Chiu & Rush, 2020) and time series from applications ranging from control systems (Franklin et al., 2002) to robotics (Doucet et al., 2009).

Furthermore, HMMs bear close relation to the widely adopted Partially Observable Markov Decision Process (POMDP) framework in reinforcement learning (e.g. Hausknecht & Stone, 2015; Rashid et al., 2020), as an HMM can be regarded as a simplification of POMDP, which has no action-input control. To this end, this paper investigates the following fundamental questions through extensive empirical experiments and theoretical analysis:

1. Can Transformer effectively learn HMM models and their variants?

2. How does its performance compare to that of Recurrent Neural Network (RNN) in terms of training speed, hyperparameter tuning difficulty, and final accuracy?

3. Furthermore, when presented with an HMM sequence of a specific length, how many attention layers are required to achieve a desired level of accuracy?

We are particularly interested in the last question due to the pivotal advantage of Transformers over RNNs in long sequence modeling: the depth of the computation graph in Transformers scales linearly

with the number of layers and remains (almost) independent of the sequence length, whereas that of RNNs scales linearly with both. It is vital to verify whether such advantage indeed exists in long sequence modeling tasks such as HMMs.

In this paper, we primarily focus on two fundamental tasks associated with HMMs: next-observation prediction and belief inference (Rabiner & Juang, 1986). Next-observation prediction involves predicting the next observation based on all preceding ones, while belief inference aims to deduce the distribution of the hidden states from previous observations. Below, we present an overview of our findings concerning the three aforementioned questions.

1. Transformers effectively learn to perform belief inference across all tested HMMs when the training dataset includes true beliefs at each time step. However, in the task of next-observation prediction, certain challenging HMM instances exist where Transformers struggle to achieve low prediction loss.

2. In comparison, RNNs demonstrate the capability to successfully tackle all tasks across the tested HMMs at a faster convergence speed, yielding lower testing error. Notably, RNN training exhibits greater robustness compared to Transformers, particularly in the realms of hyperparameter tuning and curriculum scheduling.

3. Our experiments reveal distinct patterns in the relationship between sequence length and the minimal depth required for Transformers to learn effectively. These patterns can be categorized into three groups:

   - Hard instances: There exists challenging HMM instances where Transformers struggle to learn, even for constant sequence length. These instances require further investigation to identify the underlying reasons for the learning difficulties.
   - Logarithmic scaling: For more complex sequential models such as structured HMMs, we observe an approximate logarithmic dependency between the minimal depth required and the sequence length. This relationship holds for various structured HMM instances, as corroborated by both theory and experiments.
   - Constant depth: For simple sequential models such as random HMM and linear dynamical system, a constant depth, independent of sequence length, is sufficient for Transformers to learn accurately.

4. Motivated by the hard instances, we identified two key challenging regimes of HMMs for Transformers: long mixing time and the lack of intermediate supervision signals during training. The mixing time measures how many latest observations are required to predict the next one (c.f. Section A.2), and we analyzed Transformer performance on HMMs with varying mixing times to assess the impact of mixing time. Intermediate belief states are provided for training in the belief state inference task, while only observation sequences are available in next-observation prediction task, which significantly hampers Transformer performance. Nevertheless, these factors have very little impact on RNN.

In order to address the limitations of Transformers in learning HMMs, we employ a variant of the Chain-of-Thought (CoT) (Wei et al., 2022) prompting in the training phase called block CoT. Block CoT feeds the output of the Transformer back to itself as input every $b$ tokens, which reduces to the standard CoT when $b = 1$. Our findings show that block CoT significantly decreases evaluation error and enhances the sequence length that shallow Transformers can handle.

Finally, we also complement our empirical findings by theoretical results, which proves the scaling between the sequence length and minimal depth from the perspective of expressiveness power. Specifically, it is proved that an $L$-layer finite precision Transformer is able to fit any HMMs of at least $2^L$ sequence length.

## 1.1 RELATED WORK

Our work can be viewed as part of a broader effort to assess the ability of Transformer models on simple, basic and well-defined tasks with synthetic data. Such an approach is advantageous because it allows us to precisely evaluate the Transformer's capabilities in a particular aspect, as we have access to the ground truth model that generates the training data. Below, we highlight some related works along this direction.

Recently, a line of works (e.g., Garg et al., 2022; Bai et al., 2023; Bhattamishra et al., 2023; Von Oswald et al., 2023) have studied training Transformer models for in-context learning of regression tasks (e.g., linear regressions) utilizing synthetic datasets. A notable distinction between in-context learning and learning HMMs lies in the data sequential nature. In in-context learning, all tokens within a data sequence are independently sampled from the same data distribution. Conversely, in HMMs, tokens are recursively generated, with each token strongly influenced by the preceding ones, establishing a mutual dependency among them.

Previous studies have also explored the capability of Transformer models to learn elementary algorithms, such as formal language transduction (Delétang et al., 2022), arithmetic calculation (Dziri et al., 2023), recognizing formal languages (Bhattamishra et al., 2020), sorting (Zhou et al., 2023) and learning semi-automata (Liu et al., 2022). These problems, as noted in (Liu et al., 2022), can all be considered special cases of learning the state sequence of finite-state deterministic automata, which is in turn special cases of of HMMs as noted in Appendix A.3. In comparison, our work focuses on training Transformer models to learn both the state and observation sequence of stochastic HMMs, which is a special case of more general stochastic POMDP but without input control. By focusing on stochastic HMMs, our work aims to contribute to the understanding of how Transformer models can learn and generalize from sequential data in the presence of intrinsic uncertainty.

In the belief state inference task, we provide true belief state at each step as intermediate supervision signal for the networks, while only the next observation is provided in the other task. The difference between these two tasks is very similar to training with CoT or without CoT—the belief states can be regarded as intermediate CoT signals. It is observed in previous papers that CoT greatly improves the performance of LLMs on multi-step reasoning tasks such as text generation (Wei et al., 2022; Wang et al., 2022), code generation (Li et al., 2023), multi-step computations (Nye et al., 2021), and math induction (Shao et al., 2024). There are also theoretical works Feng et al. (2023); Li et al. (2024) showing the gap of expressive power of Transformers with and without CoT.

## 2 PRELIMINARIES

In this section, we briefly introduce the basics of HMM models and neural network models considered in this paper.

### 2.1 SEQUENTIAL MODELS

An HMM can be formulated by a tuple $(\mathcal{S}, \mathcal{O}, \mathbb{P}, \mathbb{O}, S_0)$, where $\mathcal{S}$ is the state space, $\mathcal{O}$ is the observation space, $\mathbb{P}(s' \mid s)$ is the transition probability of transitioning to state $s'$ from state $s$, $\mathbb{O}(o \mid s)$ is the probability of emitting observation $o$ from state $s$, and $S_0$ is the initial state. By interacting with the HMM for $T$ steps, one can obtain a trajectory (i.e., a sequence of states and observations) $(s_0 = S_0, o_0, ..., s_T, o_T)$, where $o_t$ is sampled from distribution $\mathbb{O}(\cdot \mid s_t)$ and unobserved. $s_{t+1}$ is sampled from $\mathbb{P}(\cdot \mid s_t)$. We are particularly interested in two basic tasks for learning an HMM in this paper, which are also known as two of the three key problems of HMMs (Rabiner & Juang, 1986):

- *Next-Observation prediction:* A fundamental task is to predict the distributions of the next observation given the history of observations: $\Pr(o_{t+1} \mid o_1, ..., o_t)$.

- *Belief state inference:* Assuming the size of the state space is $n$ (i.e., $\mathcal{S} = [n]$ w.l.o.g.) for a given HMM, belief state inference aims at computing the belief state $\boldsymbol{b}_t \in \mathbb{R}^n = \Pr(s_t \mid o_1, o_2, ..., o_t)$ at step $t$ given an observation sequence $(o_1, o_2, ..., o_t)$. We provide $\boldsymbol{b}_t$ as supervision signal for each step $t$ in the training, but the network cannot use it as input since the transition is unknown.

Throughout the paper, we use $n$ to denote the size of the state space $\mathcal{S}$ if it is finite, or the dimension of $\mathcal{S}$ if it is a Euclidean space. The size of the observation space is always finite, which we denote by $m$.

## 2.2 Neural Network Models

Two fundamental sequence-to-sequence models are considered in this paper: the recurrent neural network (RNN) and Transformer.

**RNN.** Given a length-$T$ sequence $(x_1, ..., x_T)$ as input, an RNN with embedding dimension $d$ and initial hidden state $h_0 \in \mathbb{R}^d$ processes the input sequence as follows:

$$h_t = \text{ReLU}\left(W_1 x_t + W_2 h_{t-1} + b\right), \tag{2.1}$$

where $W_1, W_2, b$ are the parameters of the RNN.

The final output sequence is obtained by applying a linear decoder layer on sequence $(h_1, ..., h_T)$ at each position.

**Transformer.** The Transformer (Vaswani et al., 2017) is also a well-known sequence-to-sequence model with significant successes on various prediction tasks. We use a pre-LN Transformer (c.f. Appendix A.1) with depth $L$ (i.e., $L$ layers) that processes the data as follows:

Let $\boldsymbol{X}^{(0)} \in \mathbb{R}^{T \times d}$ be the output of a position-wise embedding layer ($d$ is the embedding dimension of the Transformer) given $m_0$-dimensional length-$T$ input sequence $(x_1, x_2, ..., x_T)^\top \in \mathbb{R}^{T \times m_0}$, the Transformer apply $L$ attention blocks sequentially on $\boldsymbol{X}^{(0)}$. The $l$-th attention block transforms the input by

$$\boldsymbol{Y}^{(l-1)} = \boldsymbol{X}^{(l-1)} + \text{Attn}^{(l)}\left(\boldsymbol{X}^{(l-1)}\right), \boldsymbol{X}^{(l)} = \boldsymbol{Y}^{(l-1)} + \text{FFN}^{(l)}\left(\boldsymbol{Y}^{(l-1)}\right), \quad l \in [L], \tag{2.2}$$

where $\text{Attn}$ is a multi-head self-attention layer and $\text{FFN}$ is a two-layer feed-forward network. The final output is obtained by forwarding $\boldsymbol{X}^{(L)}$ to a linear readout layer.

## 3 Model

In this section, we introduce the HMMs and their variant models explored in this paper, broadly classified into two categories: fast-mixing models and slow-mixing or non-mixing structured models. The mixing speed characterizes the "effective length" of past histories that influence the current belief state. For instance, in a fast-mixing model, the belief state at the current step is essentially influenced only by a few of the most recent observations, making them more amenable to fitting by neural networks. The motivation for studying these HMMs is discussed in Appendix A.5.

### 3.1 Fast-Mixing HMMs

**RanHMM: Random HMM.** The initial set of sequential models under consideration comprises random HMM instances with randomly initialized transition and emission probabilities. Our primary focus is on the belief state inference problem within these random HMMs. This choice is motivated by the observation that, as per (A.4), next-observation prediction is a relatively simpler task compared to belief state inference when dealing with **random** HMMs.

**RanLDS: Random Linear Dynamical System.** A linear dynamical system is given by the following equation $x_{t+1} = A x_t + \zeta_t, y_t = B x_t + \xi_t$, where $x_t \in \mathbb{R}^n$ is the (hidden) state at step $t$, $y_t \in \mathbb{R}^m$ is the observation at step $t$, and $\zeta_t, \xi_t$ are independent random noises. It can be regarded as a continuous HMM with linear transition and emission. We choose $A, B$ as random orthogonal matrices with $n = m$ and $\zeta_t, \xi_t$ standard Gaussian noises for simplicity. It's worth noting that predicting the belief state and the next observation is equivalent in this context, given that $B$ is orthogonal. Therefore, our focus lies on next-observation prediction, distinguishing it from the RanHMM model.

The mixing time for all HMM models are summarized in Table 1.

### 3.2 Slow or Non-mixing Models

**Cyclic-DET: Deterministic Cyclic HMM.** We begin by constructing an aperiodic HMM which never converges to any stationary distribution. Consider a deterministic Markov Decision Process

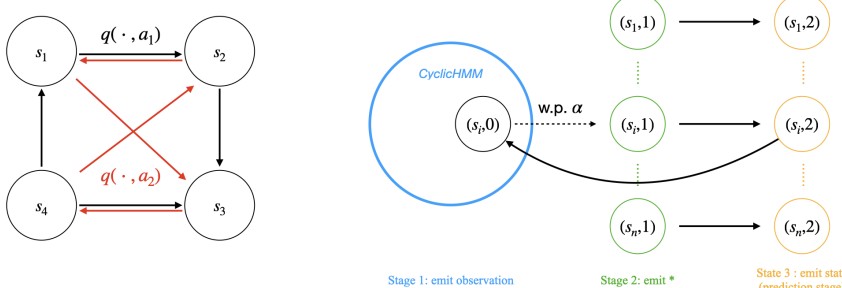

Figure 1: An illustration of `Cyclic-DET` model and `Cyclic-HARD` model. Left: A `Cyclic-DET` model with 4 states and 2 actions. The transition graph of each action is a cyclic permutation over the state space. Different actions may induce different cyclic permutation. Right: Given a `Cyclic-RND` or `Cyclic-DET` model, the `Cyclic-HARD` model transforms it into a larger HMM. The transition in `Cyclic-HARD` model always goes from stage 1 to 3 then back to stage 1. The dotted line denotes a stochastic transition from stage 1 to stage 2 with probability $\alpha$, and the solid line denotes deterministic transition. States in stage 2 always emit a signal observation $*$ indicating the entrance of stage 3, and states in stage 3 emit the current state as observation.

(MDP) with $n$ states, $m$ actions, and the state transition function $q : [n] \times [m] \to [n]$. For each action $i \in [m]$, the state transition $q(\cdot, i)$ forms a cyclic permutation (i.e., a single cycle) over the state space (see the left of Figure 1 for an example). Let $b_t \in [n]$ be the state at step $t$, $a_t \in [m]$ be the action at step $t$, then the updating rule is $b_{t+1} = q(b_t, a_t) \in [n]$, where action $a_t$ is assumed to be sampled from a uniform distribution over the action space. This model is equivalent to an HMM with $nm$ states and $m$ observations (c.f. Proposition A.1).

**`Cyclic-RND`: Stochastic Cyclic HMM.** Proposition A.1 indeed presents a robust argument, suggesting that any finite state MDP has an equivalent HMM representation. This insight prompts us to introduce some level of randomness into the previous MDP to generate a stochastic cyclic HMM. However, the randomness must be carefully calibrated; otherwise, it may become a fast-mixing model, making it easy for neural networks to fit. For any cyclic permutation induced by an action $i \in [m]$, we simply introduce a small probability $\varepsilon$ for state $s$ to transit to its predecessor when taking action $i$. The transition probability to its successor $q(s, i)$ is $1 - \varepsilon$.

**`Cyclic-HARD`: Cyclic HMM with multiple stages.** The task of next-observation prediction is straightforward for the three structured HMMs introduced earlier, as the next observation always follows a uniform distribution. To investigate the difficulty of next-observation prediction in these structured models, we devise a variant of the `Cyclic-DET` model depicted in Figure 1 (right part). Consider a `Cyclic-DET` model with state space $\mathcal{S}$, observation space $\mathcal{O}$, transition probability $\mathbb{P}(s' \mid s)$, and emission probability $\mathbb{O}(o \mid s)$, we construct the `Cyclic-HARD` model as follows, given a prediction rate $0 < \alpha < 1$.

The `Cyclic-HARD` model comprises three stages from left to right, with each stage having an independent copy of state space $\mathcal{S}$. The transition and emission probabilities of states in the first stage are almost identical to the `Cyclic-DET` model, except that each state in the first stage has a small probability $\alpha$ of transitioning to the second stage. The states in the second stage always emit a prediction signal $*$ as the observation and transition to the last stage. The final stage, also called the prediction stage, has states that always emit an observation indicating the state itself and then transition back to the first stage.

Our specific interest lies in the next-observation prediction accuracy of states in the third stage, which is equivalent to predicting the state after a random length of transitions. The formal definition is provided as follows suppose the state space is $\mathcal{S} \times \{0, 1, 2\}$ and observation space is $\mathcal{S} \cup \mathcal{O} \cup \{*\}$:

- For all states $(s, 0)$ in the first stage, it transitions to $(s', 0)$ with probability $(1 - \alpha)\mathbb{P}(s' \mid s)$, and transitions to $(s, 1)$ with probability $\alpha$. It emits $o \in \mathcal{O}$ with probability $\mathbb{O}(o \mid s)$.

- For all states $(s, 1)$ in the second stage, it transits to $(s, 2)$ with probability 1 and emits $*$ with probability 1.

- For all states $(s, 2)$ in the final stage (i.e., the prediction stage), it transition to $(s, 0)$ with probability 1 and emits $s$ with probability 1.

## 3.3 An Algebraic Example

We also provide an example of the sequential models beyond standard HMMs.

**MatMul: Matrix Multiplication.** A common task in physics and math is to evaluate the state of an unknown linear dynamical system given some action or observation $o_t$ at each step $t$. With a little abuse of notations, given the state $\boldsymbol{b}_t \in \mathbb{R}^n$ at step $t$ and an observation $o_{t+1}$ at next step, the next state is recursively updated as $\boldsymbol{b}_{t+1} \stackrel{\text{def}}{=} f(o_{t+1}, \boldsymbol{b}_t)$ for some linear function $f$. Here we assume $f(o, \boldsymbol{b}) = A_o \boldsymbol{b}$ for some unknown matrix $A_o$. In order to stabilize the system , we generate orthonormal matrix $A_o$ for all $o \in \mathcal{O}$. For simplicity, the observation $o_t$ at each step follows a uniform distribution over the observation space $\mathcal{O}$. It is an algebraic task requiring the networks to handle sequential data, and also a generalization of the constructed HMMs.

## 4 Experiments

We systematically conducted experiments to assess the learnability of RNNs and Transformers across various sequential models introduced in Section 3. The results consistently highlight the superiority of RNNs over Transformers in terms of both convergence speed and evaluation accuracy across all tasks. Besides, to delve deeper into the efficiency of Transformers with varying depths, we illustrate an approximate scaling relationship between sequence length and required depth in Figure 3.

The HMM models exhibit distinct patterns in terms of scaling. For belief state inference, fast-mixing models demonstrate compatibility with constant-depth Transformers, indicating ease of learnability of these models. The scaling of structured HMMs for the belief state inference task is constrained to at most $\log T$ for a specific sequence length $T$. The most challenging task lies in predicting the next observation in `Cyclic-HARD`, where Transformers of different depths consistently struggle to fit a sequence of constant length.

## 4.1 Experimental Design

**Training and evaluation data.** For a given HMM, we initiate by generating a random instance $\mathcal{M}$. Subsequently, we roll out $N_{\text{train}} = 5 \times 10^6$ trajectories, each of length $T = 120$, forming the training dataset. In a trajectory $(s_0, o_1, s_1, ..., o_T, s_T, o_{T+1}, s_{T+1})$, the input sequence is consistently $(o_1, o_2, ..., o_T)$. The target sequence is defined as $(\boldsymbol{b}_1, \boldsymbol{b}_2, ..., \boldsymbol{b}_T)$ for belief state inference (where belief states are computed using (A.3)), or $(o_2, o_3, ..., o_{T+1})$ for next-observation prediction. All trajectories are trained in a random order within a single epoch. To ensure fair comparison among different neural network models, we keep the instance $\mathcal{M}$ fixed for a particular HMM. For evaluating trained neural networks, we generate fresh data using $\mathcal{M}$, and the reported evaluation loss is the average loss across $E = 256$ trajectories.

**Model training.** We employ a standard decoder-only Transformer with learnable positional encoding (Radford et al., 2019) for both belief state inference and next-observation prediction. The depth $L$ of the Transformer is varied from 1 to 7. The RNN model is always single-layer and takes the raw sequence as input. Both models are trained by AdamW optimizer (Loshchilov & Hutter, 2017) with the MSE loss (for `MatMul` and `LDS`) or cross entropy loss (for others). The total training epochs for both models are 100. Additional details and hyperparameters can be found in Appendix C.

**Evaluation metric.** At the end of each epoch, we roll out $E$ fresh trajectories from the instance to evaluate the neural network. Suppose the sequence predicted by neural network is $(\hat{x}_1, \hat{x}_2, .., \hat{x}_T)$, which is the predicted belief state or the predicted next observation distribution. Given the groundtruth sequence $(x_1, x_2, .., x_T)$, the evaluation loss at length $t$ is defined as $\text{el}_t \stackrel{\text{def}}{=} \|\hat{x}_t - x_t\|_p / (3 - p)$, where

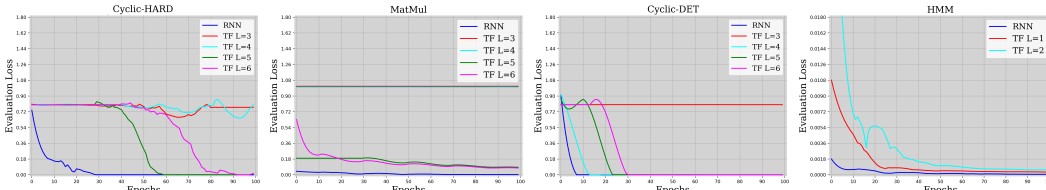

Figure 2: The evaluation loss at a specific sequence length of neural networks with optimized hyperparameter for 4 HMMs. To illustrate the difference between RNNs and Transformers of different depth, we choose the evaluation sequence length as 10, 30, 30, 120 for 4 tasks from left to right respectively.The evaluation loss of `Cyclic-HARD` model only considers the states at prediction stage since the prediction for other stages is simply a constant. The convergence speed and final accuracy of RNN are at least as good as all Transformers, which are strictly better in many cases.

$p = 2$ for `MatMul` and `RanLDS`[1], and $p = 1$ for others (because $x_t$ and $\hat{x}_t$ are distributions so it is essentially the total variation distance).

In practice, Transformers tend to learn more slowly than RNNs and are more sensitive to dataset randomness and optimization. Consequently, Transformers may not successfully fit the sequential model at the full length $T$. We consider Transformers to successfully fit the model at length $t$ with an error rate of $\epsilon$ if $\mathrm{el}_t < \epsilon$ for any $t$ starting from some epoch, where $\epsilon$ is chosen as $0.05$ or $0.1$ in the paper. The maximal length at which Transformers successfully fit at an error rate of $\epsilon$ is also referred to as the $\epsilon$-fit length.

## 4.2 CURRICULUM TRAINING

Given that fitting long sequences of length $T$ directly from scratch might pose challenges for Transformers, curriculum learning emerges as a widely adopted strategy to expedite and stabilize training (Spitkovsky et al., 2010; Wang et al., 2021; Garg et al., 2022). Curriculum learning involves dividing the training dataset into different subsets (curriculum stages) based on the measure of difficulty, and regularly switching the training data subset from simple ones to challenging ones.

In HMM models, a natural difficulty measure is the length of the training sequences (Spitkovsky et al., 2010; Garg et al., 2022). Following this curriculum design, we regularly increase the training sequence length until it reaches $T$. Motivated by the theoretical insight that an $L$-layer Transformer has a fit length of at least $2^L$ for HMMs (c.f. Section 5.2), we adopt a doubling curriculum schedule. Commencing from length $2^L$, this curriculum schedule doubles the length of the training sequence after a fixed number of epochs. The total number of curriculum stages is set to $8 - L$. The 7-layer Transformer is supposed to have only one stage at $T = 120$, the 6-layer Transformer has two stages at length 64 and 120, etc.

## 4.3 EXPERIMENTAL RESULTS

**Comparisons between RNNs and Transformers.** The comparison between RNNs and Transformers involves assessing the convergence rate, evaluation accuracy, and fit length, as depicted in Figure 2. We select four HMMs representing different categories and showcase the evaluation loss at **specific** sequence lengths for different neural networks with carefully chosen hyperparameters (c.f. Section C.1) during training. Across all four tasks, RNN consistently converges faster than all Transformers. The evaluation accuracy of RNN is consistently superior or as small as that of Transformers at all steps during training. Consequently, the fit length of RNN is at least as long as that of Transformers. We do NOT plot the evaluation loss **at the full length** $T$ because most shallow Transformers struggle to fit length $T$, whereas RNN achieves a 0.05-fit length of $T$ across all tasks.

**Scaling between fit length and depth.** Since shallow Transformers cannot fit all the HMMs at full length $T$, we provide an illustration of the scaling between fit length and depth for different

---

[1]The norm of $\|x_t\|$ is increasing in `RanLDS` model, so we compute the relative loss to be $\|\hat{x}_t - x_t\|_2 / \max(1, \|x_t\|_2)$ in this case.

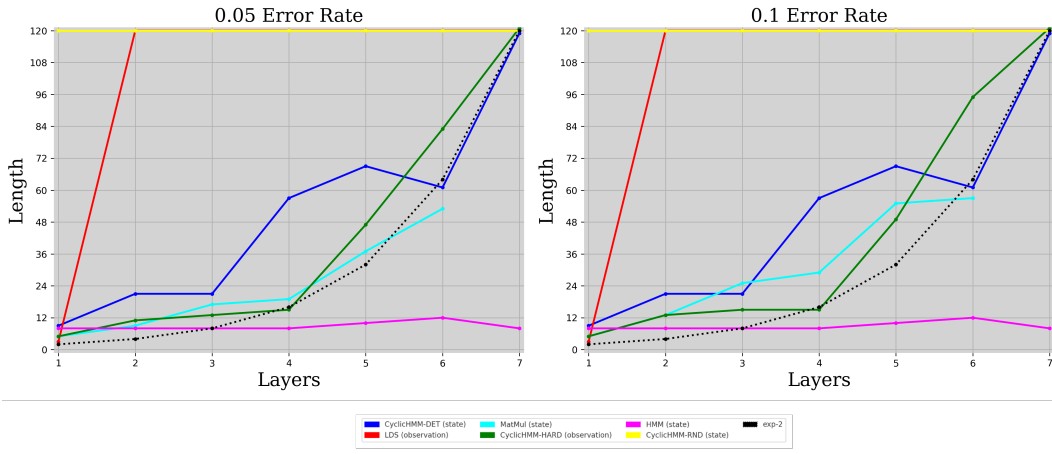

Figure 3: The approximate scaling between fit length of error rate 0.05/0.1 and the depth of the Transformer. "State" denotes a belief state inference task, and "observation" denotes a next-observation prediction task. There are roughly 3 different scaling pattern for the tasks, which is affected by the mixing time and hidden information in training data. The closeness between 0.05-fit length and 0.1-fit length reflects a small possibility of optimization caveats in the curves.

Transformers in Figure 3 (reported as the best value after hyperparameter tuning). The left two figures show the fit length of error rate 0.05 and 0.1 for all tasks. The scaling curves reveal that tasks can be roughly categorized into three classes based on Transformer performance. Fast-mixing tasks `RanHMM` and `RanLDS` can be learned by constant-depth Transformers. The most challenging `Cyclic-HARD` task cannot be fitted by an $L(\leq 7)$-layer Transformer even at constant length, while other tasks exhibit at least an exponential dependency between fit length and depth. The exponential $2^L$ scaling, as evident in the figure, aligns with our theoretical constructions discussed in the next section[2]. It is also observed during training that the Transformers suffer from optimization instability occasionally, which results in several decreasing trends on the curve.

**The impact of the mixing-time.** In order to verify the performance of Transformers on HMMs with different mixing-time, we trained a 4-layer Transformer on the `Cyclic-RND` model with different backward probability $\epsilon$ (c.f. Section 3.2). The evaluation loss at the end of training is shown in Figure 4. As $\epsilon$ increases, the mixing-time is decreasing from Table 1. Predicting the states of middle steps will be difficult since it still requires a long history to decode, while the states of late steps are easier to predict since they have converged to a fixed stationary distribution.

**The benefits of curriculum training.** In practice, we observed that the double schedule curriculum training proves beneficial in terms of training time, convergence speed, and fit length. If the number of curriculum stages is denoted as $C$, the double schedule effectively reduces the training time by a multiplicative factor proportional to $1/C$. This is because the training time scales quadratically with the sequence length. Additionally, it facilitates faster convergence and/or longer fit length for certain Transformers, as demonstrated in Figure 6 in the Appendix.

### 4.4 BLOCK CHAIN-OF-THOUGHT

For those tasks cannot be fitted by constant-depth Transformers, we investigate the possibility of applying Chain-of-Thought (CoT) (Wei et al., 2022; Feng et al., 2023) or scratchpad training (Nye et al., 2021) to reduce the required depth of the Transformers at a cost of extra training time. Intuitively, it works for the tasks that is accessible to sufficient hidden information, such as the hidden belief states

---

[2]The only exception is the result of 7-layer Transformer on `MatMul` task. The fit length of this task is still below 26 even if we have tries various tricks (e.g., different curriculum schedule, more training epochs, etc.). We conjecture this is because the training of `MatMul` task converges slower than others (c.f. Appendix C.4).

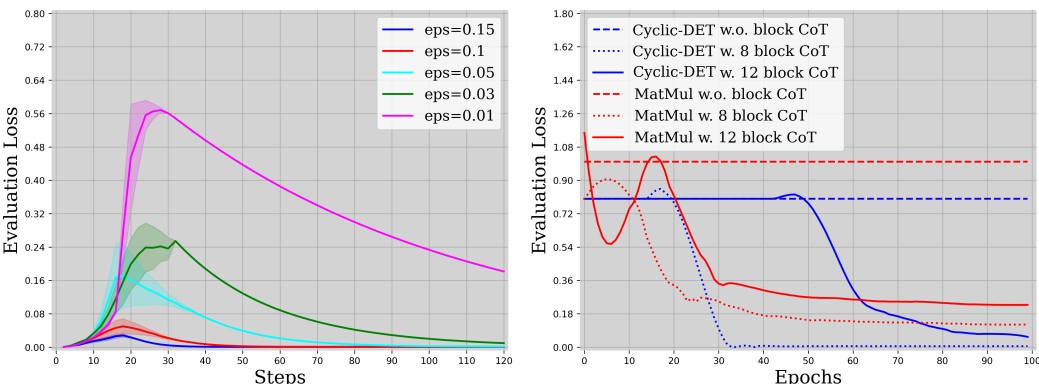

Figure 4: Left: The final evaluation loss of `Cyclic-RND` model with different $\epsilon$ and different mixing time, leading to different patterns for trained Transformers. Right: The evaluation loss at length 60 for 8/12 block CoT training for 3-layer Transformer on `Cyclic-DET` task and 4-layer Transformer on `MatMul` task. None of them has fit length 60 (dashed curves) without block CoT. The evaluation loss reduces dramatically with 8/12 block CoT, where 8/12 is approximately the half of their 0.05-fit length.

or informative observations (i.e., observations helpful to infer the hidden state) like `Cyclic-DET` model.

CoT training involves using the output of the Transformer at each step as the input to predict the next token autoregressively. However, this approach can be highly inefficient[3]. Leveraging the scaling law of fit length for layers, we can feed the output (with stop-gradient during training) back into the Transformer every $b$ steps, where $b > 1$ is a constant whose value can be guided by the fit length of the Transformer. We refer to this as $b$-block CoT training, which reduces the forward passes to $1/b$ of the original CoT training. $b$-block CoT training significantly reduces the evaluation loss of shallow Transformers at long sequence length, as demonstrated at the right side of Figure 3.

## 5 THE EXPRESSIVENESS POWER OF RNN AND TRANSFORMER

In order to understand the experimental results (e.g., the strong inductive bias of RNN, the logarithmic scaling of the fit length of Transformers, the hardness of `Cyclic-HARD` model, etc.), we ask the question whether there exists an RNN or Transformer that can produce (or express) these sequences (e.g., the belief state or observation distributions) in a theoretical perspective. This question has already been answered in general since it is known the RNN and Transformer can approximate a large number of sequence-to-sequence models (Schäfer & Zimmermann, 2006; Yun et al., 2019) given sufficient depth or width. In contrast, our interest lies in the *representation efficiency* of the neural networks to approximate the sequential models. This prompts the question of how large the neural networks should be to effectively approximate them. The proofs of all the theorems in this section can be found in Appendix E.

### 5.1 RNN

First of all, we show that RNNs can approximate HMMs with determinsitic transition conveniently.

**Theorem 5.1.** *For a deterministic HMM with state space size $n$ and observation space size $m$, there exists a single layer RNN model with embedding dimension $d = O(nm)$ and ReLU activation approximating the belief state sequence of length $T$ with no error. The $\ell_\infty$ norm of the parameters of the RNN model is bounded by $O(\|\boldsymbol{b}_0\|_2)$.*

---

[3]While we could use the hidden belief state in training labels as the autoregressive input to avoid computing the output of the Transformer, this becomes challenging when sufficient hidden information is lacking in reality.

## 5.2 TRANSFORMER

To avoid the unrealistic assumption that the neurons in Transformers have infinite precision (Dehghani et al., 2018; Pérez et al., 2019), we require the neurons to be floating-point numbers of finite precision in this paper. All the floating-point number computations will be rounded back (e.g. round-up, round-down, round-to-the-nearest) to a floating-point number as in a classical computer. To be specific, if we restrict the floating-point to have $O(\log T)$ bits, it can represent all real numbers of magnitude $O(\text{poly}(T))$ with a $O(\text{poly}(1/T))$ rounding error. After carefully analyzing the error propagation, we come to the following theorem in terms of the expressiveness of the Transformers for HMMs with deterministic transitions:

**Theorem 5.2.** *For an HMM model with deterministic transition matrix, state space size $n$, and observation space size $m$, there exists a $\log T$-precision Transformer approximating the belief state sequence of length $T$ with $O(\text{poly}(1/T))$ $\ell_\infty$ approximation error. The Transformer has depth $L = \lceil \log_2 T \rceil$, embedding dimension $2n^2 + 6$, MLP width $4n^3 + 2n^2$, and $H = 2$ attention heads. The magnitude of the parameters in the Transformer is bounded by $O(\text{poly}(T))$.*

*Proof Sketch.* For an HMM with deterministic transition, the belief state $\boldsymbol{b}_t$ is always a one-hot vector. Thus, we can reduce the HMM to a `MatMul` model with state dimension $n$ and observation space size $m$ by setting $A_o = \mathbb{P}$ for all $o \in \mathcal{O}$. It suffices to consider how to approximate the `MatMul` model. Let $A_{i:j} \overset{\text{def}}{=} \prod_{k=i}^{j} A_{o_k}$, then the output sequence of the Transformer should be $(A_{1:i}\boldsymbol{b}_0)_{i=1}^{T}$. The intuition to produce such sequence with $L$ layers is the divide-and-conquer approach (Liu et al., 2022) computing the matrix multiplications

$$A_{\max(1, j-2^l):j} = A_{\max(j-2^l, 1):\max(j-2^{l-1}, 0)} \times A_{\max(j-2^{l-1}+1, 1):j} \tag{5.1}$$

as the output of layer $l$ at position $j$, where $A_{1:0} \overset{\text{def}}{=} \boldsymbol{I}$. The analysis of the error propagation is somehow more complicated here than other divide-and-conquer construction due to the continuous representation of the state. The full proof can be found in Appendix E.2. $\square$

Approximating stochastic HMMs is more challenging since they cannot be reduced to a `MatMul` model due to the $\ell_1$ normalization step (c.f. Eqn. (A.3)). We show that if the Transformers have $T$-precision (i.e., $O(T)$ bits) and an MLP at the end of the last attention block in place of the linear DECODER layer, then it is possible for them to approximate stochastic HMMs with constant stochastic transition matrix and emission matrix:

**Theorem 5.3.** *For an HMM model with state space size $n$ and observation space size $m$ whose entries in transition matrix and emission matrix are uniformly lower bounded by $\sqrt{c_l}$, there exists a $T$-precision Transformer approximating the belief state sequence of length $T$ followed by an MLP with $O(\log T)$ layers and $O(n)$ width. The $\ell_\infty$ approximation error of the neural network is $O(\exp(-T))$. The Transformer has depth $L = \lceil \log_2 T \rceil$, embedding dimension $2n^2 + 6$, MLP width $4n^3 + 2n^2$, and $H = 2$ attention heads.*

*Remark* 5.4. Since any automaton can be formulated as an HMM (c.f., Appendix A.3), the hardness results in Liu et al. (2022) also apply to our cases. That means there exists no $\log$-precision Transformers with depth independent of $T$ and width polynomial in $T$ that can approximate any HMMs with $O(1/\text{poly}(T))$ error, unless $\text{TC}^0 = \text{NC}^1$ [4] (Feng et al., 2023; Liu et al., 2022).

## 6 CONCLUSION

We study the effectiveness of Transformers in learning HMM models and its variants from both theoretical and empirical perspective. Structured HMM models with different mixing speed are constructed to assess the accuracy of belief state inference or next-observation prediction by Transformers. We found a consistent underperformance of Transformers compared with RNN on all HMMs, and even challenging HMMs that Transformers fail to learn but RNN can successfully fit. Intuitively speaking, successful learning requires the HMM model to have either fast mixing speed or sufficient supervision signal during the learning process. We also illustrated an approximate logarithmic dependency between depth and fit length from both experiments and theory and tried the block CoT technique to overcome the limitations.

---

[4]Both are circuit complexity classes. Their relationship is widely conjectured to be $\text{TC}^0 \subset \text{NC}^1$.

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

## A  ADDITIONAL BACKGROUND AND DISCUSSION

### A.1  THE TRANSFORMER ARCHITECTURE

Different from the standard encoder-decoder Transformer architecture introduced by Vaswani et al. (2017), we employ the decoder-only Transformer in this paper to investigate its learnability of the sequential models. The decoder-only Transformers are largely applied on sequential text generations tasks, especially in large language models such as GPT-2 (Radford et al., 2019), GPT-4, LaMDA (Thoppilan et al., 2022), LLaMA (Touvron et al., 2023), etc. Let $\boldsymbol{X}_{\text{input}} \in \mathbb{R}^{T \times m}$ be the input sequence to the Transformer, where $T$ is the sequence length and $m$ is the input token dimension. The first block of the Transformer model is a position-wise linear encoder layer ENCODER : $\mathbb{R}^m \to \mathbb{R}^d$ mapping each token in $\boldsymbol{X}_{\text{input}}$ from $\mathbb{R}^m$ to $\mathbb{R}^d$, where $d$ is the embedding dimension of the Transformer. Let $\boldsymbol{X}^{(0)} \in \mathbb{R}^{T \times d}$ be the output of the linear encoder layer, it is then forwarded into $L$ attention blocks sequentially.

Each attention block consists of a self-attention layer with $H$ attention head and a two-layer MLP GeLU activated MLP. An implicit requirement for $H$ is that $H$ is a divisor of $d$ (Vaswani et al., 2017). Let the input of the $l$-th attention block be $\boldsymbol{X}^{(l-1)} \in \mathbb{R}^{T \times d}$, it propagates through a self attention layer $\text{Attn}^{(l)}$ at first, where

$$\text{Attn}^{(l)}(\boldsymbol{X}) = \text{Concat}\left(\left\{\text{softmax}\left(\boldsymbol{X}\boldsymbol{W}_Q^{(l,h)}\left(\boldsymbol{X}\boldsymbol{W}_K^{(l,h)}\right)^\top + \boldsymbol{M}\right)\boldsymbol{X}\boldsymbol{W}_V^{(l,h)}\boldsymbol{W}_O^{(l,h)}\right\}_{h=1}^H\right). \tag{A.1}$$

Here $\boldsymbol{M} \in \{0, -\infty\}^{T \times T}$ is the causal mask matrix, which is defined as $\boldsymbol{M}_{ij} = -\infty$ iff $i < j$. In other words, the output of the self-attention layer is obtained by concatenating the outputs of all the attention heads, where $\boldsymbol{W}_Q^{(l,h)}, \boldsymbol{W}_K^{(l,h)}, \boldsymbol{W}_V^{(l,h)} \in \mathbb{R}^{d \times d}$ and $\boldsymbol{W}_O^{(l,h)} \in \mathbb{R}^{d \times (d/H)}$ are the query, key, value, and output matrix respectively. The output to the self-attention layer is linked with input $\boldsymbol{X}^{(l-1)}$ by a residual connection (He et al., 2016) $\boldsymbol{Y}^{(l-1)} = \boldsymbol{X}^{(l-1)} + \text{Attn}^{(l)}(\boldsymbol{X}^{(l-1)})$. After the self-attention layer, $\boldsymbol{Y}^{(l-1)}$ is then forwarded into a 2-layer feedforward network (MLP) with a residual connection as the output of the attention block:

$$\text{FFN}^{(l)}(\boldsymbol{X}) = \sigma\left(\boldsymbol{X}\boldsymbol{W}_1^{(l)}\right)\boldsymbol{W}_2^{(l)}, \boldsymbol{X}^{(l)} = \boldsymbol{Y}^{(l-1)} + \text{FFN}^{(l)}\left(\boldsymbol{Y}^{(l-1)}\right). \tag{A.2}$$

The final output sequence of the transformer is obtained by feeding $\boldsymbol{X}^{(L)} \in \mathbb{R}^{T \times d}$ into a position-wise linear decoder layer DECODER.

We also add two LayerNorm layer right before the multi-head attention and the MLP, and feed the final output to a LayerNorm layer as suggested by a pre-LN architecture (Baevski & Auli, 2018; Wang et al., 2019; Xiong et al., 2020). The positional encoding is a learnable $d$-dim vector of length 256.

### A.2  THE MIXING TIME

Different from Markov chains, we define the mixing time of an HMM $(\mathcal{S}, \mathcal{O}, \mathbb{P}, \mathbb{O}, S_0)$ as follows in order to measure its difficulty in belief state inference and next-observation prediction tasks:

$$T_{\text{mix}} \stackrel{\text{def}}{=} \min_t \left\{ \mathbb{E}_{o_0,...,o_t} \max_{\mu_1,\mu_2} \|\Pr(S_t \mid o_0, ..., o_t, \mu_1) - \Pr(S_t \mid o_0, ..., o_t, \mu_2)\|_{\text{TV}} \leq \frac{1}{10} \right\},$$

where $\mu_1, \mu_2$ are two arbitrary initial distributions.

In the case of linear dynamical system, it is well known that $\mathbb{E}[x_t] = \zeta^r \mathbb{E}[x_{t-r}] + g_0(y_{t-r+1:t})$ for some function $g_0$ and $0 < \zeta < 1$ when the system is controllable and observable (Kalman, 1960). Therefore, we define the "mixing time" (i.e., the length of the lookback window that dominates $\mathbb{E}[x_t]$) to be $\ln(0.05)/\ln(\zeta)$ w.l.o.g.

We estimate the mixing time of the constructed HMMs in Section 3 with 1M random trajectories and 5 seeds, which are summarized in the following table.

| Model | RanHMM | RanLDS | CR-0.01 | CR-0.03 | CR-0.05 | CR-0.1 | CR-0.15 |
|-------|--------|--------|---------|---------|---------|--------|---------|
| Mixing Time | 1.6 | 3.12 | 120 | 69.4 | 42.2 | 21.4 | 14.2 |

Table 1: Average mixing time of the HMM. The CR-$\epsilon$ denotes a `Cyclic-RND` model with given backward probability $\epsilon$. The `Cyclic-DET` and `Cyclic-HARD` model do not have a stationary distribution.

### A.3 THE EQUIVALENCE BETWEEN MDP AND HMM

Consider a Markov decision process (MDP) with $n$ states and $m$ actions, let their state space and action space be $[n]$ and $[m]$ respectively. The following proposition shows that the MDP is equivalent to some HMM assuming the action $a_t$ (note that the action of the MDP is the observation of the HMM) is uniformly chosen at random for each step $t$ by the following construction:

- The state space consists of all pairs $(s, o) \in [n] \times [m]$, while the observation space is the action space $[m]$ of the MDP.

- The transition probability is defined as $\mathbb{P}((s', o') \mid (s, o)) \overset{\text{def}}{=} P_o(s', s)/m$, where $P_o(s', s)$ is the probability of transiting to state $s'$ from state $s$ given action $o$.

- The emission probability is defined as $\mathbb{O}(o' \mid (s, o)) \overset{\text{def}}{=} \mathbf{1}[o' = o]$.

**Proposition A.1.** *Given a For any $t \geq 0$, the sampling probability of any trajectory $(s_0, o_1, s_1, ..., o_t, s_t)$ is identical for the constructed HMM and the MDP.*

*Proof.* Given any $(s_0, o_1, ..., o_t, s_t)$, the probability that next observation $o_{t+1}$ equals $o$ in the HMM is

$$\sum_{s'} \frac{P_{o_t}(s', s_t)}{m} = \frac{1}{m}.$$

Therefore, the next observation distribution is a uniform distribution. On the other hand, the next state $s_{t+1}$ follows the distribution $P_{o_t}(\cdot, s_t)$ according to the construction. Therefore, these two models are equivalent for any $t$ by induction. $\square$

As deterministic automata are also MDPs (Liu et al., 2022), the construction also applies for them.

### A.4 THE BELIEF STATE INFERENCE TASK AND NEXT OBSERVATION PREDICTION TASK

**Belief state inference.** Assuming the size of the state space is $n$ (i.e., $|\mathcal{S}| = n$, the states are numbered from 1 to $n$) for a given HMM, belief state inference aims at computing the belief state $b_t \in \mathbb{R}^n$ at step $t$ given an observation sequence $(o_1, o_2, ..., o_t)$. The belief state $b_t$ is defined as the posterior probability of the HMM being at each state given the observation sequence $(o_1, o_2, ..., o_t)$: $\boldsymbol{b}_t(s) \overset{\text{def}}{=} \Pr(s_t = s \mid o_1, o_2, ..., o_t)$. It is easy to derive the following equation using Bayes' rules

$$\boldsymbol{b}_{t+1} = \frac{\text{diag}(\mathbb{O}(o_{t+1} \mid \cdot))\mathbb{P}\boldsymbol{b}_t}{\|\text{diag}(\mathbb{O}(o_{t+1} \mid \cdot))\mathbb{P}\boldsymbol{b}_t\|_1}, \tag{A.3}$$

where $\mathbb{O}(o \mid \cdot) = (\mathbb{O}(o \mid 1), ..., \mathbb{O}(o \mid n))^\top \in \mathbb{R}^n$, $\text{diag}(v)$ is the diagonal matrix generated by vector $v$, and $\mathbb{P} \in \mathbb{R}^{n \times n}$ is the transition matrix with $\mathbb{P}(s', s) = \mathbb{P}(s' \mid s)$ with a little abuse of notations.

**Next-Observation prediction.** Another fundamental task is to predict the distributions of the next observation given the history of observations. Given belief state $\boldsymbol{b}_t$ we have

$$\Pr(o_{t+1} \mid o_1, ..., o_t) = \mathbb{O}\mathbb{P}\boldsymbol{b}_t. \tag{A.4}$$

However, it is more realistic in the sense that the observations are convenient to access for an HMM with unknown transition and emission probability, while the belief states are not.

### A.5 MOTIVATIONS FOR THE CONSTRUCTED HMMS

We also dicuss our motivation for studying the HMMs and why we construct the HMMs in the main text.

**Why studying HMM models?** First, the investigation into the learnability of Transformers on various HMMs holds intrinsic value due to the universality of HMM models. This is pointed out in the introduction as the HMMs serve as useful tools for a wide range of practical problems such as part-of-speech (Kupiec, 1992) and named-entity recognition (Zhou & Su, 2002) in NLP, time-series forecasting (Zhang et al., 2019).

Furthermore, many real-world problems in control systems and reinforcement learning can be abstracted into HMMs as their simplest instances. Understanding the capabilities and limitations of Transformers in learning these models provides crucial insights that extend beyond HMMs themselves. For instance, the partially observable Markov decision process (POMDP) model, which extends HMMs by incorporating actions at each step, is a cornerstone in reinforcement learning. POMDPs are typically used to model plenty of complex sequential decision-making tasks such as robot navigation, fault detection, video game AI, etc. By investigating how Transformers perform on HMMs, we pave the way for understanding their efficacy in tackling more complex problems like POMDPs (since the HMMs are special cases of POMDPs). This is an important problem given the abundance of research efforts aimed at devising efficient reinforcement learning algorithms for learning POMDPs and their applications in various domains (see, e..g, Nguyen et al. (2020) for a survey of methods, and Cassandra (1998) for a survey of applications).

**Why constructing the specific HMMs in the main text?** The HMMs can be divided into two types: random instances and structured instances. The `RanHMM` and `RanLDS` are random instances chosen as they represent a natural starting point for exploring Transformers' learning capabilities on HMMs. Notably, the successful learning of these random instances by constant-layer Transformers suggests that HMMs lacking specific structures are relatively easy for Transformers to learn, which is because random instances have a very small mixing time (c.f. Appendix A.2).

It then inspires us to study how the mixing time of the HMM related to the performance of the Transformers. Therefore, we construct the aperiodic HMM instance `Cyclic-DET`, which requires the Transformers to consider all previous tokens to predict the next token, instead of only checking the latest ones as in the random instances (i.e., it has a long credit assignment length). The `Cyclic-RND` model further enables us to adjust the mixing time of the HMMs and verify the scaling between mixing time and the performance of HMMs.

In order to study the difference between belief state inference and next observation prediction tasks on non-mixing models, we constructed the `Cyclic-HARD` model. The core difference between `Cyclic-HARD` model and `Cyclic-DET` model is that belief state inference provides intermediate belief state as supervision signal while next observation prediction does not. The results show that it is crucial for the Transformers to have intermediate supervision signals.

Lastly, we also construct a generalization of HMMs for the application in physics or math–the `MatMul` model. The results align well with the HMMs.

# B  DETAILS OF EXAMPLES OF THE CONSTRUCTED HMM MODELS

In this section, we provide additional details and a running example based on Figure 1 of the `Cyclic-DET`, `Cyclic-RND`, `Cyclic-HARD` model constructed in Section 3.

According to Figure 1, the core design of the `Cyclic-DET` is a 4-state MDP $\mathcal{M}$ with 2 actions (Left of Figure 1). This MDP $\mathcal{M}$ can be transformed into an HMM $\mathcal{H}$ with 8 states and 2 observations according to Proposition A.1. The states of $\mathcal{H}$ are $\{(s_i, a_k)|i \in \{1, 2, 3, 4\}, k \in \{1, 2\}\}$, where $s_i$ are the states of $\mathcal{M}$ and $a_k$ are the actions of $\mathcal{M}$. With a little abuse of notation, we denote the observation space of $\mathcal{H}$ to be $\{a_1, a_2\}$, the same as the action space of $\mathcal{M}$.

Denote the initial belief state of $\mathcal{H}$ by $\boldsymbol{b}_0$, where $\boldsymbol{b}_0((s_1, a_1)) = \boldsymbol{b}_0((s_1, a_2)) = 1/2$. Since the hidden state $(s_i, a_k)$ always emits the observation $a_k$, the first observation $o_0$ emitted by the initial state can be $a_1, a_2$ with equal probability. Assume $o_0 = a_1$ (resp. $o_0 = a_2$), the next state in $\mathcal{H}$ must be $(s_2, a_1)$ or $(s_2, a_2)$ (resp. $(s_3, a_1)$ or $(s_3, a_2)$) according to the belief state update formula (A.3). The probability of these two states are both equal to 1/2, and the emission probability of $o_1$ is also a uniform distribution over $\{a_1, a_2\}$. Following this procedure, it is easy to observe that the state distribution of $\mathcal{M}$ is always equal to the marginal distribution of $s_i$ over states of $\mathcal{H}$.

The only difference for the `Cyclic-RND` model is that the MDP for the `Cyclic-RND` model adds a slightly "backward" probability given a state and an action (c.f. Section 3.2). It can be shown that the state distribution of the MDP is still the same as the marginal distribution of $s_i$ of the HMM.

Now we explain the HMM `Cyclic-HARD`. The first stage of the `Cyclic-HARD` model is exactly a `Cyclic-DET` model except for the $\alpha$-probability of entering stage 2. Suppose the `Cyclic-HARD` model has run $t$ steps without entering stage 2 (i.e., the `Cyclic-DET` model has run $t$ steps) with observation sequence $o_0, ..., o_t$, then the state $s_t$ can be decoded from taking action $o_0, ..., o_t$ in order on $\mathcal{M}$. The state distribution of the `Cyclic-DET` model is $\boldsymbol{b}_t((s_t, a_1)) = \boldsymbol{b}_t((s_t, a_2)) = 1/2$. According to the construction of the `Cyclic-HARD` model, the state distribution of `Cyclic-HARD` model is $\boldsymbol{b}_t((s_t, a_1, 0)) = \boldsymbol{b}_t((s_t, a_2, 0)) = 1/2$.

Then the `Cyclic-HARD` model enters stage 2, with the next state distribution being $\boldsymbol{b}_{t+1}((s_t, a_1, 1)) = \boldsymbol{b}_{t+1}((s_t, a_2, 1)) = 1/2$ since the state $s_t$ does not change. The observation $o_{t+1}$ is a special character * to notify the entrance of stage 3. Afterwards, it enters stage 3 with $\boldsymbol{b}_{t+2}((s_t, a_1, 2)) = \boldsymbol{b}_{t+2}((s_t, a_2, 2)) = 1/2$ and a deterministic observation $s_t$ at this step. The learner is asked to predict the observation $s_t$ for the observation prediction task.

## C  DETAILS OF EXPERIMENTS

**Training.** The training data consists of $N_{\text{train}} = 5 \times 10^6$ trajectories rolled out from the same random instance for each task. We change $N_{\text{train}}$ to $10^6$ for the simplest task `RanHMM`, `RanLDS`, and the block CoT training to save the computation time. The input data is the observation sequence $(o_0, ..., o_T)$ of length $T + 1$ ($o_0$ is a placeholder symbol) concatenated by a 3-dim positional encoding $(\sin(\pi t/4T), \cos(\pi t/4T), 1)$ at position $t$. The target data is the belief state of length $T + 1$ for belief state filtering tasks or next observation sequence of length $T$ for next-observation prediction tasks. In epoch $l$, the training loss is computed as

$$\frac{1}{T_l} \cdot \sum_{t=0}^{T_l} \mathcal{L}\left(\hat{x}_t, x_t\right), \tag{C.1}$$

where $\hat{x}_t$ is the output of the neural network given $(o_0, ..., o_t)$, $x_t$ is the training label, $\mathcal{L}$ is the loss function, and $T_l \leq T$ is the training sequence length at epoch $l$. $\mathcal{L}$ is chosen as the MSE loss for `MatMul` and `RanLDS`, and chosen as the cross entropy loss for other tasks. The training sequence length $T_l$ is $T$ if curriculum training is disabled, and set according to the curriculum stage if it is toggled on.

**Tasks.** Although the combination of belief state filtering problem and observation prediction problem with each HMM is possible, many of them are trivial. For example, predicting the next observation distribution in `MatMul`, `Cyclic-DET`, and `Cyclic-RND` is trivial since it follows a uniform distribution. Generally speaking, predicting the belief state is harder than predicting the next observation distribution since the latter is a linear mapping of the former. Importantly, we assume the access to belief states (should be hidden) at each step as training labels in the belief state filtering problem, but the next-observation prediction problem does not have access to the hidden belief states. Therefore, the belief state filtering problem in the paper provide much more hidden information during training than observation prediction. For the `Cyclic-HARD` model, we use `Cyclic-DET` to construct the first stage of `Cyclic-HARD` instead of `Cyclic-RND` to reduce the number of states. The instance used for training and evaluation is randomly generated but keep fixed for all the neural networks. The state dimension (or number of states if it is discrete) and observation dimension (or number of observations if it is discrete) are both 5 for `Cyclic-DET`, `MatMul`, `RanLDS`, `RanHMM` model, and constructed accordingly for `Cyclic-RND` and `Cyclic-HARD` model. The initial state is the first state for all tasks. We choose the parameter $\alpha = 1/T$ for `Cyclic-HARD` and $\varepsilon = 0.01$ for `Cyclic-RND` model.

**Evaluation.** The evaluation stage is at the end of each epoch. $E = 256$ trajectories are rolled out freshly, and the neural networks are fed in the sequences to do the prediction. The evaluation loss is at step $t$ is $\|\hat{x}_t - x_t\|_p/(3 - p)$ where $p = 2$ for `MatMul` and `RanLDS`, and $p = 1$ for others (i.e., total variation distance). The reported evaluation loss is the average over $E$ trajectories.

**Curriculum Training.** We employ a double schedule for curriculum training for Transformers (no curriculum training for RNNs). For an $l$-layer Transformer, we choose $8 - l$ curriculum stages and set the length of each stage to be $\lfloor 100/(8 - l) \rfloor$ epochs. The training sequence length of the first stage is $2^l$, and doubled immediately after $\lfloor 100/(8 - l) \rfloor$ epochs.

**Block Chain-of-Thought.** The $b$ block CoT training feeds the output of the Transformer back into it every $b$ steps. For the belief state filtering tasks, the output is the predicted belief state at current step. Therefore, the necessary computation depth is reduced from $T$ to $b$ if the prediction is approximately correct. For the observation prediction task, the output is the distribution of the next observation conditioned on current observation sequence $\mathbb{E}[o_{t+1} \mid o_0, ..., o_t]$. This conditional distribution may be highly correlated with the hidden belief state $\boldsymbol{b}_t$ (such as the prediction stage of `Cyclic-HARD`), or be irrelevant with the hidden belief state (such as the `Cyclic-RND` model). A measurement of such "correlation" is called the observability of the HMM (Golowich et al., 2022). The block CoT also works for observation prediction task with good observability (i.e., the correlation is strong), since the hidden state can be (approximately) inferred from the observation distribution. Since the observability of the first stage of `Cyclic-HARD` model is very bad, the `Cyclic-HARD` model cannot be resolved by block CoT training.

Theoretically speaking, the value of $b$ can be determined by the fit length of the Transformers, if the error rate is ignored. In reality, the prediction error at early steps will accumulate to later steps and so we have to choose a smaller value than $\epsilon$-fit length if we hope to reduce the evalution loss below $\epsilon$. We choose the half of the $\epsilon$-fit length in our experiments.

The averaged training time of different block sizes on `MatMul` and `Cyclic-DET` are listed in Table 2. The time cost of $b$-block CoT is approximately $1/b$ of that of vanilla CoT.

|  | block length 1 | block length 8 | block length 12 | no block CoT |
|---|---|---|---|---|
| `MatMul` | 4838 | 608 | 390 | 94 |
| `Cyclic-DET` | 4828 | 620 | 393 | 94 |

Table 2: Training time (in seconds) per epoch of block CoT for different tasks of length 60 on 4 GPUs. We choose the 3-layer Transformer for both tasks.

## C.1 Hyperparameters and Packages

The RNN models is employed from pytorch directly with embedding dimension 64, and number of layers 1. The Transformer model has embedding dimension 512, number of heads 8, MLP layer 2 with GeLU activation, and drop out rate 0.1. The optimizer is AdamW (Loshchilov & Hutter, 2017) with default parameters in pytorch. The initial learning rate for both models are `1e-3`, and decays by a factor of 0.5 every 20 epochs. We adopt the learning rate warmup procedure (Xiong et al., 2020) to warmup the learning rate from `1e-7` to `1e-3` with a linear increasing for 4000 optimization steps. The batch size is chosen from 256, 512, or 1024 depending on the layers of the Transformers, and 256 for RNN.

We also conducted ablation study to determine the best hyperparameters (e.g., the embedding dimension, the number of heads) of the Transformers and reduce variance of the training process. A brief illustration in terms of the embedding dimension and the number of heads is shown in Figure 5. The curves are drawn from multiple seeds and multiple layers with shaded area as 95% confidence interval. We choose the best configuration for the remaining of the tasks.

## C.2 Benefits of Curriculum Training

Besides saving lost of training time, curriculum training also helps warm-up the model, so as to accelerate the training as well as the final performance. Figure 6 shows the convergence speed and final value of fit length would be better with curriculum training.

## C.3 The Possibility of Overfitting

In order to verify whether the large Transformers overfit the training data, we listed the training loss at the end of the last epoch of Transformers and RNNs in Table 3 ("w. CT" means with curriculum

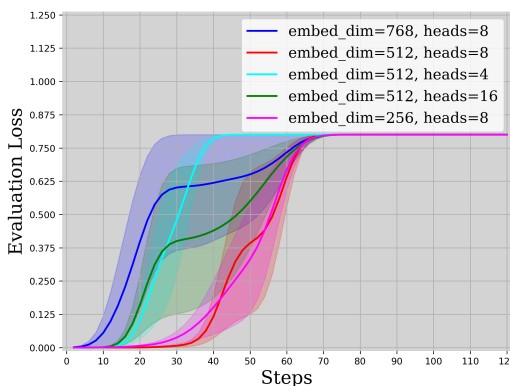

Figure 5: The evaluation loss at the end of training for different embedding dimension and number of heads for Transformers.

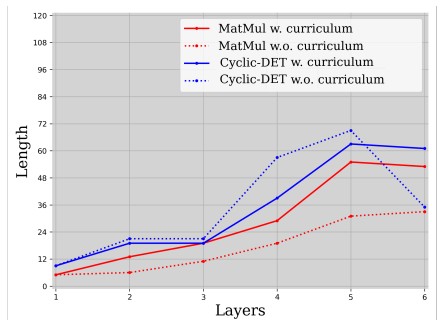
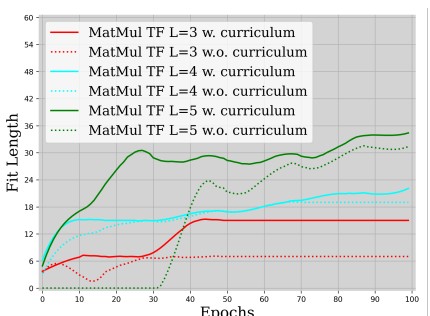

Figure 6: The benefits of curriculum training. Left: The $0.1$-fit length of Transformers with different depth, reported as the best of 2 experiments with different seed. The fit length of curriculum training is comparable with vanilla training in `Cyclic-DET` in general, and better than vanilla training in `MatMul`. Right: The convergence speed for different Transformers on `MatMul` model. The convergence of $0.1$-fit length for curriculum training is consistently faster than vanilla training.

training). From the table, we know the fitting ability of Transformers on the `MatMul`, `Cyclic-DET`, and `Cyclic-RND` model is much worse than RNNs due to the sequential nature of HMMs. For the fast-mixing models `RanHMM` and `RanLDS`, the performance of the RNNs and Transformers are comparable since the predicting these models only require a short memory.

### C.4    THE FAILURE CASE OF 7-LAYER TRANSFORMERS

We conjecture the failure of a 7-layer Transformer to fit the `MatMul` task primarily due to some optimization issues, and it requires a longer training process to tackle the issue. It can be observed from the experiments that the training loss converges much slower for `MatMul` than other tasks from Table 4. We have tried various tricks such as different curriculum scheduling, smaller training dataset, different warmup epochs, different learning rate, but they all fail to address the issue.

### C.5    IMPLEMENTATIONS AND RESOURCES

The RNN and Transformer are implemented with `Pytorch` from scratch. Each of the experiments are trained on 4 `NVIDIA GeForce RTX 4090` GPUs for 2-20 hours, where a single worker runs on each GPU.

| Task | Model | Final training loss |
|------|-------|---------------------|
| MatMul | RNN | $2.3475 \times 10^{-6}$ |
| | TF ($L = 5$) | 0.10927 |
| | TF ($L = 5$), w. CT | 0.09885 |
| | TF ($L = 6$) | 0.11073 |
| | TF ($L = 6$), w. CT | 0.10999 |
| Cyclic-DET | RNN | $2.2662 \times 10^{-7}$ |
| | TF ($L = 5$) | 0.67059 |
| | TF ($L = 5$), w. CT | 0.76278 |
| | TF ($L = 6$) | 0.68182 |
| | TF ($L = 6$), w. CT | 0.77835 |
| Cyclic-RND | RNN | $3.7769 \times 10^{-6}$ |
| | TF ($L = 5$) | 1.24707 |
| | TF ($L = 6$) | 1.13770 |
| RanHMM | RNN | 1.40139 |
| | TF ($L = 2$) | 1.40107 |
| RanLDS | RNN | 2.58242 |
| | TF ($L = 2$) | 2.61545 |

Table 3: The final training loss of different models on different tasks.

| Epoch | 60 | 70 | 80 | 90 | 100 | 125 | 150 | 175 | 200 |
|-------|-----|-----|-----|-----|-----|-----|-----|-----|-----|
| Cyclic-DET | 0.014 | 0.006 | 0.004 | 0.005 | 0.003 | N/A | N/A | N/A | N/A |
| Cyclic-RND | 1.506 | 1.503 | 1.503 | 1.502 | 1.503 | N/A | N/A | N/A | N/A |
| MatMul | 0.1069 | 0.0999 | 0.0969 | 0.0934 | 0.0916 | 0.0894 | 0.0885 | 0.0875 | 0.0870 |

Table 4: The training loss of 3 tasks of 7-layer Transformers. The total epochs for MatMul is increased to 200. The training for Cyclic-DET and Cyclic-RND have already converged at the end of epoch 100 (note that the loss for Cyclic-RND is larger due to the randomness and we implement an augmented HMM constructed in Appendix A.3). The training for MatMul converges much slower than the other two tasks.

## D  TECHNICAL LEMMAS

### D.1  USEFUL LEMMAS FOR MLP

First of all, it is ensured that an MLP with GeLU activation can approximate a scalar product.

**Lemma D.1** (Lemma C.1 of Feng et al. (2023))**.** *Let $f : \mathbb{R}^2 \to \mathbb{R}$ be a two-layer MLP with GeLU activation with 4 hidden neurons at the second layer. For any $\epsilon > 0$ and $M > 0$, there exist a set parameters of $f$ such that $|f(a, b) - ab| \le \epsilon$ for all $a, b \in [-M, M]$. The $\ell_\infty$ norm of the parameters are bounded by $O(\text{poly}(M, 1/\epsilon))$.*

It is straightforward to show that a two-layer MLP can simulate a matrix multiplication with Lemma D.1.

**Lemma D.2.** *Given two matrices $A, B \in \mathbb{R}^{n \times n}$, let the vector $\text{vec}(X) \in \mathbb{R}^{n^2}$ be the vectorization of matrix $X$. There exists a two-layer MLP $g : \mathbb{R}^{2n^2+1} \to \mathbb{R}^{n^2}$ with $4n^3$ hidden neurons and GeLU activation such that for any input vector $[\text{vec}(A), \text{vec}(B), 1] \in \mathbb{R}^{2n^2}$ with $\|A\|_F \le M - n$, $\|B\|_F \le M$, it holds that $\|g([\text{vec}(A), \text{vec}(B)]) - \text{vec}((A - \boldsymbol{I})B)\|_\infty \le \epsilon$. The $\ell_\infty$ norm of the parameters are bounded by $O(\text{poly}(M, n, 1/\epsilon))$.*

*Proof.* According to the matrix multiplication formula, it suffices to compute all the scalar product $(A - \boldsymbol{I})_{ik} B_{kj}$ for $1 \le i, j, k, \le n$ at the hidden layer, the total number of which is $n^3$. The output layer can be used to gather these products to the desired output $\text{vec}((A - \boldsymbol{I})B)$. For a positive number $\epsilon' > 0$, the constructed MLP $f$ in Lemma D.1 is

$$f(a, b) = \frac{\sqrt{2\pi}\lambda^2}{8} \left( \sigma\left(\frac{a+b}{\lambda}\right) + \sigma\left(\frac{-a-b}{\lambda}\right) - \sigma\left(\frac{a-b}{\lambda}\right) - \sigma\left(\frac{-a+b}{\lambda}\right) \right), \qquad \text{(D.1)}$$

where $0 < \lambda \lesssim M^3/\epsilon'$, and $\sigma$ is the GeLU activation.

Therefore, we can compute and store

$$u_{ijk} \stackrel{\text{def}}{=} \left( \sigma \left( \frac{(A - \boldsymbol{I})_{ik} + B_{kj}}{\lambda'} \right), \sigma \left( \frac{-(A - \boldsymbol{I})_{ik} - B_{kj}}{\lambda'} \right), \sigma \left( \frac{(A - \boldsymbol{I})_{ik} - B_{kj}}{\lambda'} \right), \sigma \left( \frac{-(A - \boldsymbol{I})_{ik} - B_{kj}}{\lambda'} \right) \right) \in \mathbb{R}^4$$

using the $4n^3$ hidden neurons of $g$ for all $i, j, k$.

By choosing an appropriate $\lambda' = O(\text{poly}(M, 1/\epsilon'))$, a single neuron at the output layer of $g$ approximate $\sum_{k=1}^n (A - \boldsymbol{I})_{ik} B_{kj}$ with $\ell_\infty$ error $n\epsilon'$ by a linear combination of all entries of $\{u_{ijk}\}_{k=1}^n$. It implies $\|g([\text{vec}(A), \text{vec}(B)]) - \text{vec}((A - \boldsymbol{I})B)\|_\infty \le n\epsilon'$. The theorem is proved by choosing $\epsilon' = \epsilon/n$. $\qquad\square$

The next lemma shows that two-layer MLPs with GeLU activation and ReLU activation are equivalent.

**Lemma D.3** (Lemma C.2 of Feng et al. (2023)). *Given $\epsilon > 0$, for any two-layer MLP $f$ with GeLU activation with parameter scale bounded by $M$, there exists a two-layer MLP $g$ with ReLU activation of the same size such that for any $x$ it holds that $\|f(x) - g(x)\|_\infty \le \epsilon$. The parameters of $g$ is bounded by $O(\text{poly}(M, 1/\epsilon))$.*

## D.2 Softmax to Approximate Hard Maximum

The following lemma quantifies the error of using softmax to approximate hard max in an $n$-dimensional vector.

**Lemma D.4** (Lemma 4 of Liu et al. (2022)). *Suppose $z \in \mathbb{R}^n$, the $\text{softmax} : \mathbb{R}^n \to \mathbb{R}^n$ function transforms $z$ into*

$$\text{softmax}(z)_i = \frac{e^{z_i}}{\sum_{j=1}^n e^{z_j}}. \tag{D.2}$$

*Let $t^* \stackrel{\text{def}}{=} \arg\max_t z_t$, and suppose for any $t \ne t^*$ it holds that $z_t \le z_{t^*} - \gamma$, then*

$$\|\text{softmax}(z) - e_{t^*}\|_1 \le 2ne^{-\gamma}. \tag{D.3}$$

## D.3 Sinusoidal Positional Encoding

**Lemma D.5.** *For any $0 \le \alpha < \alpha + \pi/4T \le \beta < \pi/4$, it holds that*

$$\cos(\alpha) - \cos(\beta) \ge \frac{\pi^2}{32T^2}. \tag{D.4}$$

*Proof.* Define $f_{\alpha,\beta}(x) \stackrel{\text{def}}{=} \cos(x) - \cos(x + \beta - \alpha)$ for $0 \le x \le \pi/4$, then $f_{\alpha,\beta}(0) \ge \pi^2/32T^2$ given that fact that $\cos x \le 1 - x^2/2$.

Since $f'_{\alpha,\beta}(x) = \sin(x + \beta - \alpha) - \sin(x) \ge 0$, it is proved that

$$f_{\alpha,\beta}(\alpha) = \cos(\alpha) - \cos(\beta) \ge f_{\alpha,\beta}(0) \ge \frac{\pi^2}{32T^2}. \tag{D.5}$$

$\qquad\square$

## D.4 Robust Matrix Multiplication

**Lemma D.6.** *Given two matrices $A, B \in \mathbb{R}^{n \times n}$ and their approximation $\widehat{A}, \widehat{B} \in \mathbb{R}^{n \times n}$ such that*

$$\| \text{vec} \left( A - \widehat{A} \right) \|_\infty \le \alpha, \| \text{vec} \left( B - \widehat{B} \right) \|_\infty \le \beta. \tag{D.6}$$

*Then it holds that*

$$\| \text{vec} \left( AB - \widehat{A}\widehat{B} \right) \|_\infty \le \alpha \left\| \widehat{B} \right\|_1 + \beta \left\| A \right\|_\infty. \tag{D.7}$$

The proof to this lemma is elementary.

# E PROOF OF MAIN THEORETICAL RESULTS

## E.1 PROOF OF THEOREM 5.1

For any deterministic HMM $(\mathcal{S}, \mathcal{O}, \mathbb{P}, \mathbb{O}, S_0)$, the belief state $\boldsymbol{b}_t$ at any step $t \geq 0$ is guaranteed to be a one-hot vector. Therefore, it almost reduces to an `MatMul` model with $A_o \stackrel{\text{def}}{=} \mathbb{P}$ for all $o \in \mathcal{O}$. The only difference is that $A_o$ is now a deterministic transition matrix instead of an orthogonal matrix, which is negligible to the approximation of the `MatMul` model for both RNNs and Transformers (because $A_o$ keeps the $\ell_2$ norm of an one-hot vector). The state dimension of this `MatMul` model is $n$, and the observation space size of it is $m$. Now we show how to approximate the output of an `MatMul` model.

Recall the updating rule of `MatMul` model (c.f. Section 3.3):

$$s_{t+1} = A_{o_{t+1}} s_t. \tag{E.1}$$

$s_0$ is fixed and $A_o A_o^\top = I$ for any $o \in \mathcal{O}$.

The RNN updating rule is

$$h_t = \text{ReLU}\left(W_i x_t + b_i + W_h h_{t-1} + b_h\right). \tag{E.2}$$

Define the following sequence $\bar{h}_t \in \mathbb{R}^n$ such that

$$\bar{h}_{t+1} = \sum_{o=1}^m \text{ReLU}\left(A_o \bar{h}_t + \alpha \mathbf{1} e_o^\top e_{o_{t+1}} - \frac{\alpha}{2}\mathbf{1}\right) - \frac{\alpha}{2}\mathbf{1}, \tag{E.3}$$

where $\bar{h}_0 = s_0$ and $\alpha > 2 \max_{o \in \mathcal{O}, 1 \leq t \leq T} \|A_o \bar{h}_t\|_\infty$. We will prove that $\bar{h}_t = s_t$ for all $0 \leq t \leq T$.

Leveraging a inductive argument, suppose it is known that $s_t = \bar{h}_t$. It holds that

$$\text{ReLU}\left(A_o \bar{h}_t + \alpha \mathbf{1} e_o^\top e_{o_{t+1}} - \frac{\alpha}{2}\mathbf{1}\right) = \begin{cases} A_o \bar{h}_t + \alpha \mathbf{1}/2 & \text{if } o = o_{t+1} \\ \mathbf{0} & \text{otherwise.} \end{cases} \tag{E.4}$$

Then $\bar{h}_{t+1} = s_{t+1}$ due to $\bar{h}_t = s_t$.

Next we construct an RNN that implements (E.3) with $d = nm$. Let $h_t \in \mathbb{R}^d$ be the hidden state at step $t$, we write $h_t$ as

$$h_t = \left[h_{t,1}^\top, h_{t,2}^\top, ..., h_{t,m}^\top\right]^\top, \tag{E.5}$$

where $h_{t,i} \in \mathbb{R}^n$ for any $t, i$. Suppose for step $t \geq 1$ we have

$$h_{t,i} = \text{ReLU}\left(A_i \bar{h}_{t-1} + \alpha \mathbf{1} e_i^\top e_{o_t} - \frac{\alpha}{2}\mathbf{1}\right) \tag{E.6}$$

and as inductive hypothesis. Since $\bar{h}_t = \sum_{o=1}^m h_{t,o} - \alpha \mathbf{1}/2$, then it is straightforward to construct weight matrix $W_h \in \mathbb{R}^{d \times d}$ and bias $b_h \in \mathbb{R}^d$ so that

$$W_h h_t + b_h = \left[A_1 \bar{h}_t, A_2 \bar{h}_t, ..., A_m \bar{h}_t\right]. \tag{E.7}$$

Moreover, we choose $W_i \in \mathbb{R}^{d \times n}$ and $b_i \in \mathbb{R}^d$ such that

$$W_i x_{t+1} + b_i = \alpha \left[\mathbf{1} e_1^\top e_{o_{t+1}}, \mathbf{1} e_2^\top e_{o_{t+1}} - \frac{\alpha}{2}, ..., \mathbf{1} e_m^\top e_{o_{t+1}}\right] - \frac{\alpha}{2}\mathbf{1}^d \tag{E.8}$$

since $x_{t+1} = e_{o_{t+1}}$. Therefore, it holds that

$$h_{t+1,i} = \text{ReLU}\left(A_i \bar{h}_t + \alpha \mathbf{1} e_i^\top e_{o_{t+1}} - \frac{\alpha}{2}\mathbf{1}\right), \tag{E.9}$$

which indicates that (E.6) holds for any $t \geq 1$ as long as it holds for $t = 1$.

When $t = 1$, we shall construct $h_0$ so that (E.6) holds for $t = 1$ with the constructed parameters by (E.7) and (E.8), which is true as long as $\bar{h}_0 = \sum_{o=1}^m h_{0,o} - \alpha \mathbf{1}/2$. Therefore, we can simply choose $h_0$ as

$$h_0 = \left[s_0 - \frac{\alpha}{2}\mathbf{1}^n, \mathbf{0}^n, ..., \mathbf{0}^n\right]. \tag{E.10}$$

The only remaining problem now is to determine the value of $\alpha$. Since $\|A_o \bar{h}_t\|_\infty \leq \|A_o \bar{h}_t\|_2 = \|A_o s_t\|_2 = \|s_t\|_2 = \|s_0\|_2$, it suffices to choose $\alpha = 4\|s_0\|_2$.

### E.2 PROOF OF THEOREM 5.2

It suffices to consider an `MatMul` model with state dimension $n$ and observation space size $m$ according to the first paragraph of E.1.

Denote the constructed $L$-layer (decoder-only) Transformer as $\mathcal{T}$, which has embedding dim $d = 2n^2 + 6$, (2-layer) MLP width $4n^3 + 2n^2$, and $H = 2$ attention heads. Let us recall the forwarding architecture of $\mathcal{T}$. Given the (one-hot) input sequence $(o_1, o_2, ..., o_T)^\top \in \mathbb{R}^{T \times m}$, we first transforms it into an augmented input sequence $\boldsymbol{X}_{\text{input}} \in \mathbb{R}^{(T+1) \times (m+4)}$ before feeding it into $\mathcal{T}$. This is achieved by adding an extra token to the input from the beginning modifying the input sequence to be $(o_0, o_1, ..., o_T)^\top \in \mathbb{R}^{(T+1) \times (m+1)}$, where $o_0 = \boldsymbol{e}_{m+1}$ is a specially token marking the beginning of the sequence. Motivated by Liu et al. (2022), we concatenate a 3-dimensional sinusoidal positional encoding at each position to form the $\boldsymbol{X}_{\text{input}}$ as

$$\boldsymbol{X}_{\text{input}} = \begin{bmatrix} o_0^\top, \sin\left(\frac{\pi \cdot 0}{4T}\right), \cos\left(\frac{\pi \cdot 0}{4T}\right), 1 \\ \vdots \\ o_i^\top, \sin\left(\frac{\pi i}{4T}\right), \cos\left(\frac{\pi i}{4T}\right), 1 \\ \vdots \\ o_T^\top, \sin\left(\frac{\pi}{4}\right), \cos\left(\frac{\pi}{4}\right), 1 \end{bmatrix}. \tag{E.11}$$

In the rest of the proof, we will give a brief summary of construction at first, and leave the technical details as the last part of the proof.

**Brief construction.** Feeding $\boldsymbol{X}_{\text{input}}$ into the $\mathcal{T}$, it will be transformed into an embedding sequence $\boldsymbol{X}^{(0)} \in \mathbb{R}^{(T+1) \times d}$ by a linear encoding layer, where $d$ is the embedding dimension of $\mathcal{T}$. Choosing $d = 2n^2 + 6$, the encoding layer is constructed so that

$$\boldsymbol{X}^{(0)} = \begin{bmatrix} \vdots \\ \boldsymbol{0}^{n^2}, 0, 0, 0, \text{vec}\left(\Lambda_{i,0}\right), \sin\left(\frac{\pi i}{4T}\right), \cos\left(\frac{\pi i}{4T}\right), 1 \\ \vdots \end{bmatrix}, \tag{E.12}$$

where $\Lambda_{i,0} \overset{\text{def}}{=} A_{o_i}$ for $0 \leq i \leq T$ and $A_{o_0} = \boldsymbol{I}_n$. Assume all the operations have **no approximation error**, we show that the output of the $l$-th ($l \geq 1$) attention block of $\mathcal{T}$ is

$$\boldsymbol{X}^{(l)} = \begin{bmatrix} \vdots \\ \boldsymbol{0}^{n^2}, 0, 0, 0, \text{vec}\left(\Lambda_{i,l}\right), \sin\left(\frac{\pi i}{4T}\right), \cos\left(\frac{\pi i}{4T}\right), 1 \\ \vdots \end{bmatrix}, \tag{E.13}$$

where $\Lambda_{i,l} \overset{\text{def}}{=} A_{\max(i-2^l+1,0):i}$ for $l \geq 0$ and $\Lambda_{i,l} \overset{\text{def}}{=} \boldsymbol{I}$ for $l < 0$. Recall that $A_{i:j} = \prod_{k=i}^{j} A_{o_k}$.

Suppose this is true for $\boldsymbol{X}^{(l-1)}$, we now prove this induction for layer $l$ assuming no approximation error. Recall the $l$-th self-attention layer processes as

$$\text{Attn}^{(l)}(\boldsymbol{X}^{(l-1)}) = \text{Concat}\left(\left\{\text{softmax}\left(\boldsymbol{X}^{(l-1)}\boldsymbol{W}_Q^{(l,h)}\left(\boldsymbol{X}^{(l-1)}\boldsymbol{W}_K^{(l,h)}\right)^\top + \boldsymbol{M}\right)\boldsymbol{X}^{(l-1)}\boldsymbol{W}_V^{(l,h)}\boldsymbol{W}_O^{(l,h)}\right\}_{h=1}^{H}\right), \tag{E.14}$$

where $\boldsymbol{W}_Q^{(l,h)}, \boldsymbol{W}_K^{(l,h)}, \boldsymbol{W}_V^{(l,h)}\boldsymbol{W}_O^{(l,h)}$ are query, key, value, output matrices of the $h$-th head.

For the first attention head $h = 1$, we construct matrix $\boldsymbol{W}_Q^{(l,1)}$ so that

$$\boldsymbol{X}^{(l-1)}\boldsymbol{W}_Q^{(l,1)} = \gamma \cdot \begin{bmatrix} \vdots \\ \sin\left(\frac{\pi(i-2^{l-1})}{4T}\right), \cos\left(\frac{\pi(i-2^{l-1})}{4T}\right) \\ \vdots \end{bmatrix} \in \mathbb{R}^{(T+1) \times 2}. \tag{E.15}$$

The $\boldsymbol{W}_K^{(l,1)}$ matrix is constructed to form

$$\boldsymbol{X}^{(l-1)}\boldsymbol{W}_K^{(l,1)} = \gamma \cdot \begin{bmatrix} \vdots \\ \sin\left(\frac{\pi i}{4T}\right), \cos\left(\frac{\pi i}{4T}\right) \\ \vdots \end{bmatrix} \in \mathbb{R}^{(T+1)\times 2}. \tag{E.16}$$

In this way, we can choose an appropriate value of $\gamma$ to ensure the attention mask matrix of the first attention head is approximately

$$\text{softmax}\left(\boldsymbol{X}^{(l-1)}\boldsymbol{W}_Q^{(l,1)}\left(\boldsymbol{X}^{(l-1)}\boldsymbol{W}_K^{(l,1)}\right)^\top + \boldsymbol{M}\right) = \begin{bmatrix} \vdots \\ \lambda_{i,l}^\top \\ \vdots \end{bmatrix} \in \mathbb{R}^{(T+1)\times(T+1)}, \tag{E.17}$$

where $\lambda_{i,l} \overset{\text{def}}{=} \boldsymbol{e}_{\max(i-2^{l-1},0)} \in \mathbb{R}^{(T+1)}$. The value of $\gamma$ is chosen as

$$\gamma = \frac{4\sqrt{2T}\log(2T/\eta)}{\pi}, \quad \text{where } \eta = \frac{1}{(8n)^{L+1}\cdot T} \tag{E.18}$$

As a result, the output of the first attention head $\boldsymbol{H}^{(l,1)} \in \mathbb{R}^{(T+1)\times(d/2)}$ will be approximately

$$\boldsymbol{H}^{(l,1)} = \begin{bmatrix} \vdots \\ \text{vec}\left(\Lambda_{\max(i-2^{l-1},0),l-1}\right), 0, 0, 0 \\ \vdots \end{bmatrix} \tag{E.19}$$

by constructing appropriate value and output matrices.

For the second head $h = 2$, we simply produce a all-zero matrix $\boldsymbol{H}^{(l,2)} = \boldsymbol{0}^{(T+1)\times(d/2)}$. The final output of the attention layer $\boldsymbol{Y}^{(l-1)} \in \mathbb{R}^{(T+1)\times d}$ is produced by concatenating the output of two attention heads plus a residual connection:

$$\boldsymbol{Y}^{(l-1)} = \boldsymbol{X}^{(l-1)} + \left[\boldsymbol{H}^{(l,1)}, \boldsymbol{H}^{(l,2)}\right] \tag{E.20}$$

$$= \begin{bmatrix} \vdots \\ \text{vec}\left(\Lambda_{\max(i-2^{l-1},0),l-1}\right), 0, 0, 0, \text{vec}\left(\Lambda_{i,l-1}\right), \sin\left(\frac{\pi i}{4T}\right), \cos\left(\frac{\pi i}{4T}\right), 1 \\ \vdots \end{bmatrix}. \tag{E.21}$$

The construction of the 2-layer MLP at the end of the $l$-th attention block will be divided into two parts. First of all, we use $4n^3$ hidden neurons to compute $(\Lambda_{\max(i-2^{l-1},0),l-1} - \boldsymbol{I}_n)\Lambda_{i,l-1} = \Lambda_{i,l} - \Lambda_{i,l-1}$ according to Lemma D.2. Then a simple 2-layer ReLU network with $2n^2$ hidden neurons can flip the sign of the first $n^2$ entries of the input $\boldsymbol{Y}^{(l-1)}$, which we can use a GeLU network with the same size to simulate according to Lemma D.3. The output of the MLP is

$$\text{FFN}^{(l)}(\boldsymbol{Y}^{(l-1)}) = \begin{bmatrix} \vdots \\ -\text{vec}\left(\Lambda_{\max(i-2^{l-1},0),l-1}\right), 0, 0, 0, \text{vec}\left(\Lambda_{i,l} - \Lambda_{i,l-1}\right), 0, 0, 0 \\ \vdots \end{bmatrix} \in \mathbb{R}^{(T+1)\times d}. \tag{E.22}$$

Finally, the output of the $l$-th attention block is

$$\boldsymbol{Y}^{(l-1)} + \text{FFN}^{(l)}(\boldsymbol{Y}^{(l-1)}) = \begin{bmatrix} \vdots \\ \boldsymbol{0}^{n^2}, 0, 0, 0, \text{vec}\left(\Lambda_{i,l}\right), \sin\left(\frac{\pi i}{4T}\right), \cos\left(\frac{\pi i}{4T}\right), 1 \\ \vdots \end{bmatrix} = \boldsymbol{X}^{(l)}, \tag{E.23}$$

which proves the induction of (E.13).

Without approximation error, the output of the last attention block $\boldsymbol{X}^{(L)}$ is

$$\boldsymbol{X}^{(L)} = \begin{bmatrix} \vdots \\ \boldsymbol{0}^{n^2}, 0, 0, 0, \operatorname{vec}\left(\Lambda_{i,L}\right), \sin\left(\frac{\pi i}{4T}\right), \cos\left(\frac{\pi i}{4T}\right), 1 \\ \vdots \end{bmatrix}, \tag{E.24}$$

where $\Lambda_{i,L} = A_{0:i}$ since $L = \lceil \log_2 T \rceil$. A final linear decoder layer transformers $\boldsymbol{X}^{(L)}$ to the desired output sequence

$$(\boldsymbol{b}_1, \boldsymbol{b}_1, ..., \boldsymbol{b}_T) = (A_{0:1}\boldsymbol{b}_0, A_{0:2}\boldsymbol{b}_0, ..., A_{0:T}\boldsymbol{b}_0). \tag{E.25}$$

**The self-attention layer.** We still assume no approximation error in the sequel. According to the expression of $\boldsymbol{X}^{(l-1)} \in \mathbb{R}^{(T+1) \times d}$ in (E.13), the query matrix $\boldsymbol{W}_Q^{(l,1)} \in \mathbb{R}^{d \times 2}$ is constructed as

$$\boldsymbol{W}_Q^{(l,1)} = \gamma \cdot \begin{bmatrix} 0 & 0 \\ \vdots & \vdots \\ \cos\left(\frac{\pi 2^{l-1}}{4T}\right) & \sin\left(\frac{\pi 2^{l-1}}{4T}\right) \\ -\sin\left(\frac{\pi 2^{l-1}}{4T}\right) & \cos\left(\frac{\pi 2^{l-1}}{4T}\right) \\ 0 & 0 \end{bmatrix} \tag{E.26}$$

according to the sine/cosine difference formula. The construction of $\boldsymbol{W}_K^{(l,1)}$ is straightforward.

As a result, the attention mask matrix is

$$\boldsymbol{M}_{ij}^{(l)} \stackrel{\text{def}}{=} \left[\boldsymbol{X}^{(l-1)}\boldsymbol{W}_Q^{(l,1)}\left(\boldsymbol{X}^{(l-1)}\boldsymbol{W}_K^{(l,h)}\right)^{\top}\right]_{ij} = \gamma^2 \cos\left(\frac{\pi(j - i + 2^{l-1})}{4T}\right) \tag{E.27}$$

for $j \leq i$. The entry for $j > i$ is obviously $-\infty$ since $\boldsymbol{M}$ is the causal mask. The $i$-th row of this matrix is

$$\gamma^2 \left[\cos\left(\frac{\pi(-i + 2^{l-1})}{4T}\right), \cos\left(\frac{\pi(1 - i + 2^{l-1})}{4T}\right), ..., \cos\left(\frac{\pi 2^{l-1}}{4T}\right), -\infty, ...\right]. \tag{E.28}$$

Since $0 \leq i, 2^{l-1} \leq T$, the maximum value is take at position $i - 2^{l-1}$ if $i - 2^{l-1} \geq 0$, and position $0$ otherwise. Choosing

$$\gamma = \frac{4\sqrt{2}T \log(2T/\eta)}{\pi}, \tag{E.29}$$

the softmax of this row would be $\lambda_{i,l} = \boldsymbol{e}_{\max(i-2^{l-1},0)}$ with $\ell_1$ error $\eta$ according to Lemma D.5 and Lemma D.4.

Finally, it suffices to use matrix $\boldsymbol{W}_V^{(l,1)} = \boldsymbol{I}$ and $\boldsymbol{W}_O^{(l,1)} \in \mathbb{R}^{d \times (d/2)}$ by

$$\boldsymbol{X}^{(l-1)}\boldsymbol{W}_V^{(l,1)}\boldsymbol{W}_Q^{(l,1)} = \begin{bmatrix} \vdots \\ \operatorname{vec}\left(\Lambda_{i,l-1}\right), 0, 0, 0 \\ \vdots \end{bmatrix} \tag{E.30}$$

to produce the output $\boldsymbol{H}^{(l,1)}$ of the first attention head.

**The approximation error.** The approximation error of $\mathcal{T}$ needs to be bounded carefully in order to prove the $O(\operatorname{poly}(1/T))$ total $\ell_\infty$ error due to the exponential propagation over layers. We assume the number of states $n$ is a constant in the proof.

There are three places in the construction introducing approximation error:

- The softmax operation to the attention mask matrix (E.28). Let

$$\operatorname{softmax}\left(\widehat{\boldsymbol{M}}^{(l)}\right) \stackrel{\text{def}}{=} \begin{bmatrix} \vdots \\ \widehat{\lambda}_{i,l}^{\top} \\ \vdots \end{bmatrix} \in \mathbb{R}^{(T+1) \times (T+1)} \tag{E.31}$$

be the exact output of the softmax attention mask matrix in $\mathcal{T}$, then $\|\widehat{\lambda}_{i,l} - \lambda_{i,l}\|_1 \leq \eta$ for any $i, l$ according to Lemma D.5, D.4 and the choice of $\gamma$ (c.f. (E.18)).

- The matrix multiplication operation performed by the MLP, which is the place error may propagate over layers. Let $\widehat{\boldsymbol{X}}^{(l)}$ be the exact output of $\mathcal{T}$ after $l$-th attention block, then

$$
\widehat{\boldsymbol{X}}^{(l)} = \begin{bmatrix} \vdots \\ \widehat{\boldsymbol{\varepsilon}}_{i,l}, 0, 0, 0, \text{vec}\left(\widehat{\Lambda}_{i,l}\right), \sin\left(\frac{\pi i}{4T}\right), \cos\left(\frac{\pi i}{4T}\right), 1 \\ \vdots \end{bmatrix}. \tag{E.32}
$$

Here the matrix $\widehat{\Lambda}_{i,l}$ may not equal to $\Lambda_{i,l}$ due to the approximation error of the MLP.

- The last occurrence of the approximation error is $\widehat{\boldsymbol{\varepsilon}}_{i,l} \in \mathbb{R}^{n^2}$. This term is supposed to be $\boldsymbol{0}$ in our construction, which would be the case if our MLP uses ReLU as activation. According to Lemma D.3, it holds that $\|\widehat{\boldsymbol{\varepsilon}}_{i,l}\|_\infty \leq \eta$ for any $i, l$ as long as we use our GeLU MLP to simulate the ReLU MLP with parameters bounded by $O(\text{poly}(1/\eta)) = O(\text{poly}(T))$ according to (E.18).

Now we analyze the error propagation in a single attention block. Suppose at the beginning of the $l$-th attention block we have $\|\text{vec}(\Lambda_{i,l-1} - \widehat{\Lambda}_{i,l-1})\|_\infty \leq \epsilon_{l-1}$ for all $0 \leq i \leq T$, so $\epsilon_0 = 0$.

The first place introducing the error is the output of the first attention head $\boldsymbol{H}^{(l,1)}$ in (E.19). The exact output $\widehat{\boldsymbol{H}}^{(l,1)}$ of the Transformer $\mathcal{T}$ is

$$
\widehat{\boldsymbol{H}}^{(l,1)} = \text{softmax}\left(\widehat{\boldsymbol{M}}^{(l)}\right) \widehat{\boldsymbol{X}}^{(l-1)} \boldsymbol{W}_V^{(l,1)} = \begin{bmatrix} \vdots \\ \widehat{\lambda}_{i,l}^\top \\ \vdots \end{bmatrix} \cdot \begin{bmatrix} \vdots \\ \text{vec}\left(\widehat{\Lambda}_{i,l-1}\right), 0, 0, 0 \\ \vdots \end{bmatrix}. \tag{E.33}
$$

The approximation can then be decomposed as

$$
\widehat{\boldsymbol{H}}^{(l,1)} - \boldsymbol{H}^{(l,1)} = \text{softmax}\left(\widehat{\boldsymbol{M}}^{(l)}\right)\left(\widehat{\boldsymbol{X}}^{(l-1)}\boldsymbol{W}_V^{(l,1)} - \boldsymbol{X}^{(l-1)}\boldsymbol{W}_V^{(l,1)}\right) \tag{E.34}
$$

$$
+ \left(\text{softmax}\left(\widehat{\boldsymbol{M}}^{(l)}\right) - \text{softmax}\left(\boldsymbol{M}^{(l)}\right)\right)\boldsymbol{X}^{(l-1)}\boldsymbol{W}_V^{(l,1)}. \tag{E.35}
$$

Elementary inequalities show that

$$
\left\|\text{vec}\left(\widehat{\boldsymbol{H}}^{(l,1)} - \boldsymbol{H}^{(l,1)}\right)\right\|_\infty \leq \max_{0 \leq i \leq T}\left(\left\|\widehat{\lambda}_{i,l}\right\|_1 \cdot \max_{0 \leq j \leq T}\left\|\text{vec}\left(\widehat{\Lambda}_{j,l-1} - \Lambda_{j,l-1}\right)\right\|_\infty\right) \tag{E.36}
$$

$$
+ \max_{0 \leq i \leq T}\left(\left\|\widehat{\lambda}_{i,l} - \lambda_{i,l}\right\|_1 \cdot \max_{0 \leq j \leq T}\left\|\text{vec}\left(\Lambda_{j,l-1}\right)\right\|_\infty\right) \tag{E.37}
$$

$$
\leq (1 + \eta)\epsilon_{l-1} + \eta. \tag{E.38}
$$

The second inequality follows from the definition of $\epsilon_{l-1}$, $\|\widehat{\lambda}_{i,l} - \lambda_{i,l}\|_1 \leq \eta$, and the fact that $\Lambda_{j,l-1}$ is an orthogonal matrix.

Write $\widehat{\boldsymbol{H}}^{(l,1)}$ as

$$
\widehat{\boldsymbol{H}}^{(l,1)} = \begin{bmatrix} \vdots \\ \widehat{\boldsymbol{h}}_{i,l} \in \mathbb{R}^{n^2}, 0, 0, 0 \\ \vdots \end{bmatrix}. \tag{E.39}
$$

It is shown by (E.38) that $\|\text{vec}(\Lambda_{\max(i-2^{l-1},0),l-1}) - \widehat{\boldsymbol{h}}_{i,l}\|_\infty \leq (1 + \eta)\epsilon_{l-1} + \eta$.

The input to the MLP in $\mathcal{T}$ is

$$
\widehat{\boldsymbol{Y}}^{(l-1)} = \widehat{\boldsymbol{X}}^{(l-1)} + \left[\widehat{\boldsymbol{H}}^{(l,1)}, \boldsymbol{H}^{(l,2)}\right] = \begin{bmatrix} \vdots \\ \widehat{\boldsymbol{\varepsilon}}_{i,l-1} + \widehat{\boldsymbol{h}}_{i,l}, 0, 0, 0, \text{vec}\left(\widehat{\Lambda}_{i,l-1}\right), \sin\left(\frac{\pi i}{4T}\right), \cos\left(\frac{\pi i}{4T}\right), 1 \\ \vdots \end{bmatrix}.
$$

$$
\tag{E.40}
$$

The MLP at layer $l$ implements an approximate matrix multiplication on the matrix reshaped by $\widehat{\varepsilon}_{i,l-1} + \widehat{h}_{i,l} - I_n$ and $\widehat{\Lambda}_{i,l-1}$. Now that

$$\left\| \text{vec}\left(\Lambda_{\max(i-2^{l-1},0),l-1}\right) - \widehat{h}_{i,l} - \widehat{\varepsilon}_{i,l-1} \right\|_\infty \leq (1+\eta)\epsilon_{l-1} + 2\eta, \left\| \text{vec}\left(\Lambda_{i,l-1} - \widehat{\Lambda}_{i,l-1}\right) \right\|_\infty \leq \epsilon_{l-1} \tag{E.41}$$

we have

$$\left\| \text{vec}\left(\Lambda_{i,l} - \Lambda_{i,l-1}\right) - \text{vec}\left(\hat{\Lambda}_{i,l} - \hat{\Lambda}_{i,l-1}\right) \right\|_\infty \leq ((1+\eta)\epsilon_{l-1} + 2\eta)\left(n\epsilon_{l-1} + \sqrt{n}\right) + \sqrt{n}\epsilon_{l-1} \tag{E.42}$$

$$\leq (1+\eta)n\epsilon_{l-1}^2 + \left((1+\eta)\sqrt{n} + 2\eta n + \sqrt{n}\right)\epsilon_{l-1} + 2\eta\sqrt{n}, \tag{E.43}$$

which implies

$$\left\| \text{vec}\left(\Lambda_{i,l} - \hat{\Lambda}_{i,l}\right) \right\|_\infty \leq (1+\eta)n\epsilon_{l-1}^2 + \left((1+\eta)\sqrt{n} + 2\eta n + \sqrt{n} + 1\right)\epsilon_{l-1} + 2\eta\sqrt{n}. \tag{E.44}$$

Define $\epsilon_l \stackrel{\text{def}}{=} \max_i \|\text{vec}(\Lambda_{i,l} - \hat{\Lambda}_{i,l})\|_\infty$, then the sequence $\epsilon_l$ satisfies

$$\epsilon_l \leq (1+\eta)n\epsilon_{l-1}^2 + \left((1+\eta)\sqrt{n} + 2\eta n + \sqrt{n} + 1\right)\epsilon_{l-1} + 2\eta\sqrt{n}. \tag{E.45}$$

Thanks to the construction of $\eta$ in (E.18), we prove by induction that $\epsilon_l \leq (8n)^{l-L} \cdot T^{-1}$, which is obvious when $l = 0$. Suppose this is true for $l - 1$, then

$$\epsilon_l \leq (1+\eta)n\epsilon_{l-1}^2 + \left((1+\eta)\sqrt{n} + 2\eta n + \sqrt{n} + 1\right)\epsilon_{l-1} + 2\eta\sqrt{n} \tag{E.46}$$

$$\leq (1+\eta)n\epsilon_{l-1} + \left((1+\eta)\sqrt{n} + 2\eta n + \sqrt{n} + 1\right)\epsilon_{l-1} + 2\eta\sqrt{n} \quad \text{(from } \epsilon_{l-1} < 1) \tag{E.47}$$

$$\leq 4n(1+\eta)\epsilon_{l-1} + 2\eta n \quad \text{(from } n \geq 1) \tag{E.48}$$

$$\leq 6n\epsilon_{l-1} + 2\eta n \tag{E.49}$$

$$\leq \frac{1}{(8n)^{L-l} \cdot T}\left(\frac{3}{4} + 2\eta n T \cdot (8n)^{L-l}\right) \quad \text{(from the induction hypothesis)} \tag{E.50}$$

$$\leq \frac{1}{(8n)^{L-l} \cdot T}\left(\frac{3}{4} + \frac{1}{4}\right) \quad \text{(from the definition of } \eta) \tag{E.51}$$

$$\leq \frac{1}{(8n)^{L-l} \cdot T}. \tag{E.52}$$

This inductive argument shows that $\epsilon_L = 1/T = O(\text{poly}(1/T))$. Moreover, the parameters of $\mathcal{T}$ are all bounded by $O(\text{poly}(T,n))$ from the choice of $\eta$ and $\gamma$.

### E.3 PROOF OF THEOREM 5.3

The Transformers constructed here is slightly different from the ones constructions in Theorem 5.2 in that we choose a different $\eta$ (recall that the Transformers now has $T$-precision):

$$\eta = \frac{1}{(8n)^{L+1} \cdot \exp(4T)}.$$

Recall that $\tilde{b}_t = A_{1:t}b_0$, then from the same analysis as (E.46) to (E.52) we know that the output of the last attention block can be regrouped to vectors $\hat{b}_t$ such that $\|\hat{b}_t - \tilde{b}_t\|_\infty = O(\exp(-4T))$. It then remains to normalize the vector $\hat{b}_t$ to have 1 $\ell_1$ norm.

Next we consider how to normalize the vector $\hat{b}_t$ by a $O(\log T)$-layer MLP with width $O(n)$. The MLP is divided into two parts to achieve the following goals:

1. Find a multiple $c_0$ such that $1/2 \leq \|c_0\hat{b}_t\|_1 \leq 1$;

2. Divide the vector $c_0\hat{b}_t$ by its $\ell_1$ norm.

**Finding the multiple $c_0$.** Let $C_l \overset{\text{def}}{=} 1/c_l > 1$. On the $n$-dimensional input $\hat{\boldsymbol{b}}_t$, we calculate $\hat{\boldsymbol{b}}_{t,0} = C_l^T \hat{\boldsymbol{b}}_t, v_0 \overset{\text{def}}{=} \|\hat{\boldsymbol{b}}_{t,0}\|_1$ using the first two layers. It holds that $1 - O(\exp(-3T)) \leq v_0 \leq C_l^T + O(\exp(-3T))$ since $\|\hat{\boldsymbol{b}}_t - \tilde{\boldsymbol{b}}_t\|_\infty = O(\exp(-4T))$ and $c_l^T \leq \|\tilde{\boldsymbol{b}}_t\|_1 \leq 1$. We use

For the next $4k + 1$-st to $4k + 4$-th layers ($k = 0, 1, ..., \lceil \log_2 T \rceil$), we process the vector $\hat{\boldsymbol{b}}_{t,k}$ as follows:

- Use layer $4k + 1$ and $4k + 2$ to compute $p(v_k - C_l^{\lfloor T/2^{k+1} \rfloor})$ where

$$p(x) = \text{ReLU}\left(1 - \text{ReLU}(-x)\right) = \begin{cases} 1 & x \geq 1 \\ x & 0 \leq x < 1 \\ 0 & 0 < x \end{cases} \tag{E.53}$$

.

- Use layer $4k + 3$ and $4k + 4$ to approximate

$$v_{k+1} \overset{\text{def}}{=} p\left(v_k - C_l^{\lfloor T/2^{k+1} \rfloor}\right) c_l^{\lfloor T/2^{k+1} \rfloor} v_k + \left(1 - p(v_k - C_l^{\lfloor T/2^{k+1} \rfloor})\right) v_k \tag{E.54}$$

$$\hat{\boldsymbol{b}}_{t,k+1} \overset{\text{def}}{=} p\left(v_k - C_l^{\lfloor T/2^{k+1} \rfloor}\right) c_l^{\lfloor T/2^{k+1} \rfloor} \hat{\boldsymbol{b}}_{t,k} + \left(1 - p(v_k - C_l^{\lfloor T/2^{k+1} \rfloor})\right) \hat{\boldsymbol{b}}_{t,k} \tag{E.55}$$

by Lemma D.2 to perform the multiplication, so that $v_{k+1} = \|\hat{\boldsymbol{b}}_{t,k+1}\|_1$. We choose the parameters of these two layers to guarantee that the approximation error is $O(\exp(-3T))$. Let the output of layer $4k + 4$ for $v_k$ be $\hat{v}_k$.

Now we prove by induction that $1/2 \leq v_k \leq C_l^{\lfloor T/2^k \rfloor} + 1$ for all $k$, where the case for $k = 0$ is trivial. If $C_l^{\lfloor T/2^k \rfloor} + 2 \geq v_k \geq C_l^{\lfloor T/2^{k+1} \rfloor} + 1$, then $1 \leq v_{k+1} = v_k c_l^{\lfloor T/2^{k+1} \rfloor} \leq C_l^{\lfloor T/2^{k+1} \rfloor} + 1$. The case for $v_k \leq C_l^{\lfloor T/2^{k+1} \rfloor}$ is similar.

If $C_l^{\lfloor T/2^{k+1} \rfloor} < v_k < C_l^{\lfloor T/2^{k+1} \rfloor} + 1$, then we can rewrite $v_{k+1}$ using the difference $p_k \overset{\text{def}}{=} p(v_k - C_l^{\lfloor T/2^{k+1} \rfloor}) = v_k - C_l^{\lfloor T/2^{k+1} \rfloor}$:

$$v_{k+1} = (1 - p_k)C_l^{\lfloor T/2^{k+1} \rfloor} + p_k^2 c_l^{\lfloor T/2^{k+1} \rfloor} + p_k(2 - p_k), 0 < p_k < 1. \tag{E.56}$$

It is nor hard to show that it still holds that $1/2 \leq v_{k+1} \leq C_l^{\lfloor T/2^{k+1} \rfloor} + 1$.

Next we analyze the approximation error of multiplication. For $k = 0$ it holds that $|\hat{v}_k - v_k| = 0$. Note that

$$\hat{v}_{k+1} \overset{\text{def}}{=} p(\hat{v}_k - C_l^{\lfloor T/2^{k+1} \rfloor}) c_l^{\lfloor T/2^{k+1} \rfloor} \hat{v}_k + (1 - p(\hat{v}_k - C_l^{\lfloor T/2^{k+1} \rfloor}))\hat{v}_k.$$

Since the function $p$ is 1-Lipschitz and $v_k = O(\exp(T/2^k))$, it holds that $|\hat{v}_{k+1} - v_{k+1}| \leq O(\exp(T/2^k))|\hat{v}_k - v_k| + O(\exp(-3T))$. The $\ell_\infty$ approximation error of $\hat{\boldsymbol{b}}_{t,k}$ is the same as that of $v_k$.

Let $\bar{\boldsymbol{b}}_t$ be the output of the $4\lceil \log_2 T \rceil + 4$ layer, then it holds that $\|\bar{\boldsymbol{b}}_t/\|\bar{\boldsymbol{b}}_t\|_1 - \hat{\boldsymbol{b}}_t/\|\hat{\boldsymbol{b}}_t\|_1\|_\infty \leq O(\exp(-2T))$ and $1/4 \leq \|\bar{\boldsymbol{b}}_t\|_1 < 3/2$.

**Normalizing the vector $\bar{\boldsymbol{b}}_t$.** The output $\bar{\boldsymbol{b}}_t$ of the first part of the MLP is then fed into the second part of the MLP. Denote $0 \leq c_1 \overset{\text{def}}{=} 1 - 2\|\bar{\boldsymbol{b}}_t\|_1/3 < 5/6$, then the final target is to compute

$$\frac{2\bar{\boldsymbol{b}}_t}{3(1 - c_1)} = \frac{2\bar{\boldsymbol{b}}_t}{3}\left(1 + c_1 + c_1^2 + \cdots\right).$$

If we use $O(k)$-layers in the second part of the MLP, then we can approximate the sum

$$1 + c_1 + c_1^2 + \cdots + c_1^{2^k - 1}$$

with error $O(\exp(-T))$ by picking appropriate parameters according to Lemma D.2.

Moreover,

$$\left| \frac{1}{1 - c_1} - 1 + c_1 + c_1^2 + \cdots + c_1^{2^k - 1} \right| = \frac{c_1^{2^k}}{1 - c_1} \leq 6 \left( \frac{5}{6} \right)^{2^k}. \tag{E.57}$$

Therefore, it suffices to choose $k = O(\log T)$ to approximate $2\bar{\boldsymbol{b}}_t / 3(1 - c_1)$ in $\ell_\infty$ error $O(\exp(-T))$. Aggregating the approximation error of both parts of the MLP, the overall approximation error is $O(\exp(-T))$.

