# OpenReview forum: "On Limitation of Transformer for Learning HMMs"
_ICLR.cc/2025/Conference — Submitted to ICLR 2025_

### Official Review · Reviewer_Ttof · 2024-11-01

**Soundness:** 3
**Presentation:** 3
**Contribution:** 2
**Rating:** 5
**Confidence:** 3

**Summary:**

The authors empirically and theoretically investigate the ability of Transformers and RNNs to learn six synthetic sequential tasks derived from both random and structured Hidden Markov Models (HMMs). They justify choosing HMMs due to their broad applicability in various tasks like part-of-speech tagging, named-entity recognition, time-series forecasting, control systems, and reinforcement learning. The study introduces three key findings: (1) Transformers can effectively learn many HMMs, provided their depth increases logarithmically with sequence length; (2) RNNs perform better on certain complex HMMs and surpass Transformers in both convergence speed and accuracy; (3) Transformers struggle with long mixing times due to their lack of intermediate latent states. The authors examine the use of chain-of-thought (CoT) prompting to address these Transformer limitations.

**Strengths:**

- The choice of synthetic tasks based on random and structured HMMs is well-justified and relevant to a range of practical applications.
- The experimental setup is modern and reasonable.
- The experimental results, while expected, are interesting and theoretically supported.
- The paper is generally clear and well-structured.

**Weaknesses:**

- The implications of the empirical and theoretical findings are not discussed. For example, the authors could suggest potential improvements for Transformers in applications with long mixing times or discuss the practical applicability of RNNs, given their superior convergence speed and performance on the synthetic tasks.
- I believe the impact of the work to be limited.
- The choice of the ReLU activation in RNNs is not motivated.
- The choice of a positional encoding with a dimensionality of 3 might not be optimal.

**Questions:**

- What are the implications of your findings?
- Why was ReLU chosen over classical activations like tanh or sigmoid? Was this to address the vanishing gradient problem?
- Could you provide an ablation study on the positional encoding dimensionality? Additionally, could you conduct an experiment with RoPE on a task where the Transformer struggles (e.g., Cyclic-HARD)?
- I recommend mentioning in the main text that pre-LN was used, as it helps with convergence speed -- yet RNNs still converge faster than Transformers.
- I suggest adjusting the y-axis limit to (0, 120) in Figure 3.
- Line 68: Consider replacing "training speed" with "convergence speed" for clarity.
- Line 81: Define "LDS".
- Line 111: Replace "sequence's nature" with "sequential nature."
- Appendix A5: The motivation for the choice of tasks is great; I suggest moving (part of) it to the main text.

---

### Official Review · Reviewer_oeMU · 2024-11-02

**Soundness:** 3
**Presentation:** 2
**Contribution:** 3
**Rating:** 5
**Confidence:** 2

**Summary:**

The authors introduce various Hidden Markov models (HMM) datasets as a tool to study the learning behavior of Transformers and RNNs.  This includes RanHMM, a HMM with randomly sampled transitions and observations.  They claim that belief inference (inferring the hidden state of the HMM) is harder than predicting the emission, so they also introduce define HMMs with more structure, that they derive from Markov Decision Processes (MDPs).  This includes CyclicHMM-DET, a deterministic MDP with random actions that gives an aperiodic HMM, CyclicHMM-RND, a version that can slip backward, and Cyclic-HARD.  The demonstrate that the different model systems have quite different behaviors and mixing times.  In addition they also test two continuous varieties, MatMul and RanLDS, a random linear system.

Generally they find that RNNs learn HMMs more quickly and robustly than Transformers, and there are certain HMMs that transformers fail to learn at all.  In particular, long mixing times particularly hurt transformers as the recent past is not as predictive of the current state.

They also introduce what they call block Chain of Thought (CoT) reasoning can help transformers reduce evaluation error and learn longer sequences.  This involves feeding back the output distribution of the transformer as input.  Note that this is not just the output symbol, as they claim that this output distribution is very informative about the hidden state of the HMM.

Finally, they prove that a transformer with L layers can model a sequence of effective length 2^L, which motivates them to introduce a curriculum training by increasing the doubling the sequence length during training, which also helps.

**Strengths:**

* Understanding our tools is important in ML; different methods have different trade-offs and this is another demonstration of the types of problems that transformers struggle with and other models (or hybrid models) should be applied.
* Their proof in 5.2 of the number of layers L needed in the transformer to model an HMM of length 2^L is a useful bound that’ll find applications outside of this domain.  Many problems being solved in LLMs are highly sequential, and there can be trade-off between depth and width of each layer, for which this bound could be informative.
* Their dataset could prove interesting in understanding the behavior of other sequence models, like SSMs, and thus in developing new sequence models.

**Weaknesses:**

* HMMs are intrinsically a sequential problem - in a way, it’s unsurprising that the Transformer model would perform worse at them. E.g., expressing the “copy the symbols from N characters ago) task is pretty hard in a HMM or an RNN, but is very natural in a Transformer.  Limiting the scope of problems you look at is fine, but it would be useful to have at least a short discussion of the types of problems that would be well modeled by a HMM or not.
* The paper would have been stronger if a parallel state space model like MAMBA had been included, as it would be interesting to see parallel trained SSM could learn and represent this inherently sequential computation in a parallel manner.
* Calling their method of feeding output token distributions back in as “Chain of thought” (CoT) is a bit of loose analogy.  In CoT you prompt with other examples and the reasoning happens first, unlike in their system.  Their block CoT does appear to help, so this is more of a comment on the naming.
* Only using a single evaluation per HMM can make the results too noisy to draw reliable conclusions.
  * E.g., Figure 2, Cyclic-Hard, the fact that L=5 beat L=4 and L=6 could just be noise.  It would be better to rerun at least 3 times and plot error-bars.

**Questions:**

* How do you embed the continuous inputs and outputs?  Are the inputs/outputs of the transformer the same size as the transformer hidden state?  Similarly for the discrete
* The step/stage description in Cyclic-HARD (and corresponding Figure) I found unclear.  Could you elaborate (maybe in a short appendix) what it would actually expand into for a small problem?
  * As an aside, it would help build intuition if you provided some samples from each of the systems in an Appendix, or provided them for download.
* How do you feed the gradient through feeding in the output token in the CoT approach?  Is there a stop-gradient involved?
“network cannot use it as input since the transition is unknown.” isn’t very clear what it is referring to - it could learn the transitions after all?

---

### Official Review · Reviewer_Zf4C · 2024-11-03

**Soundness:** 3
**Presentation:** 3
**Contribution:** 3
**Rating:** 6
**Confidence:** 3

**Summary:**

This paper studies the capability of transformers in learning variants of Hidden Markov Models (HMMs), while comparing with RNNs. The authors demonstrated several key findings and support them with theoretical analysis. One such finding implies that transformers can model HMMs and its depth scales logarithmically with the input sequence length. The experiments show that there is a subset of HMMs that are challenging for transformers to learn while RNNs can model. More specifically, HMMs with long mixing time and which have limited supervision signal during training are particularly challenging for transformers but not so for RNNs. The authors show that including the output of the transformer as its input, a method the authors called block Chain of Thought, can mitigate this limitation of transformers.

**Strengths:**

The work is timely given the widespread adaptation of transformer-based models. HMMs are important area to study.

The findings are backed by theoretical analyses.

The paper is well written and easy to follow.

**Weaknesses:**

The experiments were mostly conducted with model HMMs. If experiments with real-world datasets are included, this can show this can translate to practical benefits and implications

**Questions:**

About block Chain of Thought: if the reviewer understood correctly, this is rather different from the CoT that is generally used in LLMs where instruction is given to prompt the LLM to think through the question before answering. In this paper, the block CoT refers to ‘block’ autoregressive decoding where the output predictions in the previous steps are fed as input tokens to inform generation at the current step. If that right? It might be worth clarifying, e.g. with a simple sketch diagram, to make it clearer to the reader.

Typo at Line 886: “ate” -> “at”

---

### Official Review · Reviewer_wjYr · 2024-11-05

**Soundness:** 2
**Presentation:** 2
**Contribution:** 2
**Rating:** 5
**Confidence:** 3

**Summary:**

The paper applies transformers to learning models for output sequences of several types of  HMMs and Linear Dynamical Systems models.
The authors used two main tasks for evaluation -- next observation prediction and belief state prediction from the observations.

They compared the convergence speed of the transformer models with recurrent models and found, surprisingly (at least to me)
that RNN's converged faster and better than transformer models that they implemented.

The authors explored two main factors in the learning efficiency. Firstly they explored the effect of longer mixing length
on the ability of the transformer models to learn good models. Secondly they explored the effect of uncertainty in predicting
the belief state of the model from the observations, on the predictive capacity of the models.  They found that when
the mixing length increased transformer models performed worse. They also found that when belief states were more strongly
correlated with next step predictions, then predicting belief states in the intermediate blocks could result in
improved performance by a block COT type approach.

Finally, the authors show some theoretical proofs related to -
1. The ability of RNNs and transformers to model sequences from HMMs with different levels of predictions based on depth
and length of sequences.
2. Existence of transformer models that can learn to model outputs from HMMs with arbitrary accuracy based on minimum
depth -- they used prior work which showed that transformers can model any arbitrary automata (of which HMM is but a sub-case).

**Strengths:**

I thought the idea of using sequences from HMMs to study the learning abilities of transformers quite a good one -- it is possible to control the complexity of the sequences by the Markov-order of the HMMs; it is possible to control the number of states, and output distribution etc. The authors created Cyclic-{DET, RND,HARD} models which have different properties in terms of complexity, mixing length etc.

The block COT method seems to be good way to propagate more information through the model. During training this makes the model more autoregressive, which making it less computationally demanding than a fully next step autoregressive model based of the state of the model.

**Weaknesses:**

I don't want to be too certain about what the weakness of the paper are, until I read the authors' rebuttals and clarifications, since its possible I may have missed some details and motivations.

My first reaction, is, sadly an instinctive one. In as much as it makes sense to me that deeper models should perform better, and that the more complex the sequence type, the harder it would be for an algorithm to model it. However, I am quite surprised, to see that recurrent models learn so much faster, and I want to be certain that this observation, which is repeated in various forms in the paper, holds.  I have left several questions in the Questions section to help me understand the details. What I would have to say here is that it seems to me that transformers can mimic recurrent models, by learning to put all the attention in the previous step and previous step alone. In doing so, it can cause the models to flow information almost in a recurrent way. So I am puzzled that the recurrent model can outperform it.

Another thing that comes to mind is that I think to some extent the model studies not just the learning properties of transformers, but the combination of the transformers and the loss function. HMM's can often produce complicated, multi-modal output distributions. Modeling these with an L2-loss is likely to lead to over smoothed models. Such problems are well studied in ML, back to early days of using mixture density networks, where the model's output was a mixture model, and not just a single mean value. I suppose this is true also for the Recurrent models, but have the authors explored the impact of this on the learning ? I left some clarification questions below in the questions section to tease this apart a little bit more.

It is not clear why curriculum is important here. If I understand correctly, the same HMM is used over training a model. In that case, the complexity of each task is really just dependent on the property of the HMM, and the complexity is the same through the sequence. Is it not true that a shorter sequence from the HMM should be no more complex than a longer sequence from the same HMM ?

Experimental details are somewhat sparse; specifically what were the sizes of state space, n, what were the output dimensions ? Were continuous data normalized before training transformers. For discrete outputs, what were the output space (m) ?

**Questions:**

I am afraid I had a lot of questions about the details that I think can be clarified, which will help me perform a better evaluation of the paper.  I list them here in no particular order.

1. With your RNN's -- did you end up using ReLU RNN's ? It's surprising they perform so much better than transformers, even though LSTM's were typically the best in class RNNs years past.
2. I'm not entirely sure why you created the exact forms of HMM you did in this paper. The essence of the method seems to be to generate HMMs' that have different Markov-orders and different correlations between belief states and output sequences. Could this have been explored with other alternatives such as the complexity of the Markov Chains of the hidden sequences and the complexity of the emission distributions ?
3. Can you please provide more information about the experiments ? I am not sure what was the state sizes for different experiments,
4. With discrete state HMMs, and continuous outputs it would make sense that the output distribution would be a mixture model, rather than a single model. As such using an l2 loss may cause problems.  Did the authors attempt to use mixture density network type approaches for these cases ?
5. Are the loss functions averaged over batch ? i.e. is loss divided by the batch size and sequence length ? I see only sequence length, in (B.1) and so I want to be sure.
6. B.1 lays you the loss function for training -- L2 for Matmul, RanLDS and cross entropy for other tasks. But evaluation on line 877 seems to suggest that evaluation loss is the l2 loss.. Is this also the case for categorical outputs ? I would think that the right evaluation objective would be the KL from the predictive distribution or the posterior belief state of the HMM itself. What is the case here ?
7. Do you have a study on the effect of the block size, b, of the block COT on the results ?
8. Epsilon on like 410 is overloaded with epsilon in line 344.
9. Line 455 - .."we can feed the output back into the transformer every b steps". Is this the actual output from the model ? If so, is gradient back propagated through this, and with what method (straight through, or reinforce, etc) ? If not, isn't it just teacher force model durning training, just like language models, and so training speed should be unaffected ?
10. During curriculum training, as I understand it here, you use increasingly longer sequences. Is batch size kept the same ? Are learning rate and other things adjusted for the changing batch size ? Does the training proceed from the previous point, or is learning rate warmup performed once more ?
11.  In figure 6, why does the Cyclc-DET w.o. curriculum fall so significantly when going from 5 to 6 layers ?
12. It is really detailed to follow the HMM construction -- will you be releasing the code to create Cyclic-DET and Cyclic-RND ?
13. Any thoughts on why transformers perform almost a million times worse than a single layer RNN, for Matmult, Cyclic-DET and RND -- are these are with continuous outputs ? These results are quite surprising to me.

---

### Meta-Review · Area_Chair_XAXe · 2024-12-22

**Metareview:**

## Summary
This paper examines the ability of transformers to learn sequences generated by Hidden Markov Models (HMMs). Through a combination of theoretical analysis and experiments, the authors compare transformers and Recurrent Neural Networks (RNNs) in their capacity to learn HMMs.

The experiments reveal three key findings: (1) Transformers can learn many HMMs effectively but require depth that scales logarithmically with sequence length. (2) RNNs outperform transformers in training speed and accuracy across all tested HMMs and are more capable of handling challenging HMMs. (3) Transformers face difficulties when HMMs involve long mixing times or lack access to intermediate latent states, while RNNs are less impacted.

To address these challenges, the authors propose a variant of the Chain-of-Thought (CoT) approach called “block CoT,” which improves transformers’ performance on longer sequences at the cost of increased training time. Theoretical results also demonstrate that transformers with sufficient depth can approximate HMMs accurately, building on prior findings about their expressiveness.

## Decision

Overall, this work is timely and important and the authors support their findings are backed by theoretical analyses. The paper seems to be well-written and easy to follow. However, it seems to be rushed, especially the experiments. The reviewers were not fully convinced by the claims and the experimental results. Thus, they raised several concerns during the rebuttal. However, the authors didn't submit a rebuttal to address these concerns. As a result, in its current state, this paper is not ready for acceptance.

**Additional Comments On Reviewer Discussion:**

The paper was borderline, and the reviewers raised several concerns and questions. However, the authors didn't post a rebuttal to answer them.

---

### Decision · Program_Chairs · 2025-01-22

Reject